# SEI growth on Lithium metal anodes in solid-state batteries quantified with coulometric titration time analysis

Burak Aktekin [1] ✉, Luise M. Riegger[1], Svenja-K. Otto[1], Till Fuchs[1], Anja Henss[1] & Jürgen Janek [1] ✉

Lithium-metal batteries with a solid electrolyte separator are promising for advanced battery applications, however, most electrolytes show parasitic side reactions at the low potential of lithium metal. Therefore, it is essential to understand how much (and how fast) charge is consumed in these parasitic reactions. In this study, a new electrochemical method is presented for the characterization of electrolyte side reactions occurring on active metal electrode surfaces. The viability of this new method is demonstrated in a so-called anode-free stainless steel | $Li_6PS_5Cl$ | Li cell. The method also holds promise for investigating dendritic lithium growth (and dead lithium formation), as well as for analyzing various electrolytes and current collectors. The experimental setup allows easy electrode removal for post-mortem analysis, and the SEI's heterogeneous/layered microstructure is revealed through complementary analytical techniques. We expect this method to become a valuable tool in the future for solid-state lithium metal batteries and potentially other cell chemistries.

Rechargeable batteries, particularly lithium-ion batteries (LIBs), have become a vital component of many devices we use in our daily life. It is expected that they will play a central role in electrification of transport and grid energy storage in the near future which is particularly important for the transition from fossil-based energy sources to renewable alternatives. However, a widespread transition necessitates advanced battery technologies having high specific energy and power, as well as low cost and environmental impact. For this reason, it is crucial to use electrode materials having high specific charge capacities while enabling high cell voltages at the same time, as well as materials allowing fast charge and discharge kinetics. In order to achieve this, the cathode (positive electrode) should ideally operate at the highest and the anode (negative electrode) at the lowest possible potential. Unfortunately, existing electrolyte systems have often a relatively narrow thermodynamic stability window, and thus, electrolyte decomposition reactions are observed at the surface of commonly used anodes and cathodes, e.g. graphite and $LiNi_xMn_yCo_{1-x-y}O_2$ (NMC), respectively[1].

The graphite anode has a specific charge capacity of $q_{th} = 372$ mAh g$^{-1}$ and operates at a low potential $E_H(C/Li_xC) \approx 0.1$ V vs. Li$^+$/Li. Its use in LIBs as an anode material with ethylene carbonate (EC) containing liquid carbonate electrolytes (with LiPF$_6$ salt) dates back to 1991, and today graphite-based anodes are still being used in most commercial cells[2]. Even though the carbonate-based electrolyte is not stable at such low potential, the electrolyte decomposition products form a self-passivation layer at the anode surface which is called solid electrolyte interphase (SEI)[3,4]. This self-passivation layer prevents continuous electrolyte degradation and graphite exfoliation, and thus enables the long term cyclability of LIBs. Further engineering of this protective SEI through more complex electrolyte formulations, electrolyte additives and specific formation cycles has played an important role in achieving the high-performance level of today's LIBs[5].

Commercial state-of-the-art LIBs can deliver mass specific energies of around $w_p \approx 270$ Wh kg$^{-1}$ (cell level) and alternative anode

[1]Institute of Physical Chemistry & Center for Materials Research, Justus-Liebig-Universität Giessen, D-35392 Giessen, Germany.
✉e-mail: burak.aktekin@phys.chemie.uni-giessen.de; juergen.janek@phys.chemie.uni-giessen.de

materials are sought to achieve higher specific energies ($w_p > 400$ Wh kg$^{-1}$)[6–9]. One path for increasing the specific energy is to replace graphite by silicon or lithium metal electrodes. The former has a theoretical specific charge capacity of $q_{th}(Si) \approx 3600$ mAh g$^{-1}$ while still operating at a low potential similar to graphite. However, the large volume expansion of Si during lithiation (up to 400%)[10] leads to extensive SEI damage/repair in each cycle and causes depletion of the lithium inventory of the full cell. Ideally, an even better alternative to silicon would be lithium metal since it has a specific charge capacity $q_{th} = 3860$ mAh g$^{-1}$ and electrode potential $E_H(Li^+/Li) = 0$ V (i.e. –3.04 V vs. the standard hydrogen electrode), especially if it is formed in situ in an initially lithium metal-free full cell configuration (e.g. in a so-called anode free cell). Unfortunately, problems such as extensive SEI growth also exist for lithium metal batteries (LMBs) with even an increased risk of dendritic lithium growth (and thus safety risks) in operation[11,12]. The most promising approaches to overcome these issues are the development of new electrolytes (and electrolyte additives)[13–15], the protection of the current collector (CC) and the lithium metal surface by suitable coating materials[16], and the use of architectured host structures for lithium deposition[17,18]. In determining the effectiveness of such approaches, the quantitative characterization of interfacial side reactions between the cycled lithium metal and the electrolyte (e.g. SEI growth) is of utmost importance.

All-solid-state batteries (ASSBs) are quite promising for a safer implementation of lithium metal anodes either in LMBs or in anode-free cell concepts due to the absence of flammable liquid electrolytes. High mechanical rigidity and a high lithium transference number ($t(Li^+) \approx 1$) of solid electrolytes (SEs) is potentially advantageous for mitigating the lithium dendrite growth[1] making them good candidates for batteries with lithium metal electrode. Unfortunately, most SEs are also known to have a narrow thermodynamic stability window and react with lithium metal at low potentials, thus forming either an unstable or stable SEI[19–22]. Particularly at high current densities, dendrite growth along cracks and grain boundaries is observed in inorganic SEs and this can exacerbate the interfacial side reactions—and eventually lead to short circuit and cell failure[23–28].

It is evident that a full understanding of these interfacial phenomena is crucial for a successful development of advanced battery technologies—whether they rely on a liquid, solid or hybrid electrolyte. For the measurement of the electrolyte stability window, the conventionally used electrochemical techniques are cyclic voltammetry (CV) and linear sweep voltammetry (LSV)—the latter being simply a partial (i.e. half-cycle) CV experiment. Even though these measurements can provide quite useful basic information about electrolyte stability, there are a number of pitfalls of such tests as they can overestimate the electrolyte stability window, e.g. particularly when the scan rates are high and measurement sensitivity is low, or when low surface area electrodes are used, etc[19]. In order to overcome these issues, electrolyte stability can be tested under more static conditions by staircase voltammetry[29], or under more practically relevant conditions by simulating the actual potential profile of a cell with an active electrode material working electrode (WE), e.g. synthetic charge-discharge profile voltammetry[30]. Nevertheless, since the goal in these voltammetric techniques is to study the current generated by electrolyte side reactions, redox-active electrode materials cannot be used as WEs. Instead, redox-inactive metal foils (Pt, Ni, Al, etc.) or glassy carbon are mostly used. However, it is well known that the stability of the electrolyte may be altered due to surface effects (e.g. catalytic effects) when the inert working electrode in such CV tests is replaced by the actual active electrode material[31–33]. In the case of the lithium metal anode, the active electrode material will also be consumed by side reactions affecting the local volume changes[34] near the interface. This can change the interphase microstructure and further influence the degree of side reactions. This effect could be exacerbated during lithium plating due to volume change and SEI damage associated with

the freshly deposited lithium metal, especially if dendritic lithium growth occurs[35]. Therefore, it is important to complement such voltammetric tests with other analytical methods[36,37].

In this sense, carefully performed Coulomb efficiency (CE) measurements can give useful quantitative information on side reactions on the redox-active electrode-electrolyte interface (e.g. lithium consumed during SEI formation). However, such CE measurements have their own pitfalls since cell resistance, active mass loss and dead lithium formation could contribute significantly to the observed Coulomb efficiency[38–40]. In order to differentiate and quantify the amount of charge consumed in side reactions, it is necessary to perform complementary experiments either in situ (e.g. nuclear magnetic resonance[41], synchrotron X-ray diffraction[42], etc.) or ex situ (e.g. titration gas chromatography[39,43], mass spectrometry titration[44,45], etc.).

An alternative electrochemical approach is to measure the interface resistance with electrochemical impedance spectroscopy (EIS), which can be used to estimate the surface film thickness (e.g. SEI)[22,46]. In this case, the solubility of the SEI components (in liquid electrolytes), its porosity, phase heterogeneity and the assumptions made for the electronic conductivity of SEI may induce large errors in thickness estimations. Therefore, such measurements must also be complemented by other characterization methods.

In this paper, we present a new and precise electrochemical method for the quantification of side reactions occurring between a redox-active electrode material and an (solid) electrolyte that has to the best of our knowledge not been reported before. The technique is simple, easy to perform and does not require sophisticated/expensive potentiostats yet provides highly useful information on the degree (i.e. coulombic quantification) and kinetics of side reactions. We name this method coulometric titration time analysis (CTTA), and demonstrate its application to electrodes on sulfide-type SEs ($Li_6PS_5Cl$, $Li_3PS_4$, $Li_{10}GeP_2S_{12}$) and $Li_{6.4}La_3Zr_{1.4}Ta_{0.6}O_{12}$ oxide-type SE as important model-type SEs with practical relevance. The method not only enables the quantification of side reactions by accounting for actual electrode-electrolyte interaction, but also provides a means of studying dendritic lithium growth by the selection of suitable titration parameters. Additionally, it can be utilized to analyze various electrolytes, current collectors and the effect of operation parameters such as temperature. We demonstrate that CTTA is particularly well-suited for the characterization of interfacial reactions occurring between lithium metal (and possibly also other reactive metal electrodes such as Na, Mg) and their solid electrolytes.

## Results and discussion
### Coulometric titration time analysis (CTTA)
The use of coulometry as a modern analytical technique can be traced back to early reports on the use of constant current coulometric titration[47] in 1938 and the invention of the potentiostat[48] in 1942. Since then, it has been used in a wide range of quantitative electroanalytical applications[49] such as trace analysis, e.g. in the well-known Karl Fischer coulometric titration method[50] used for the determination of the water content. In battery research, coulometric titration step based methods are commonly used to determine kinetic parameters (e.g. diffusion coefficients) of electrode active materials. Such methods consist of a series of coulometric titration steps each followed by a relaxation period. For instance, in the galvanostatic intermittent titration technique (GITT)[51], kinetic parameters are derived from the voltage response of the cell during each titration step. Alternatively, the voltage response during the early relaxation period can also be used for a similar purpose as in the case of the intermittent current interruption (ICI) technique[52]. In either case, the role of the coulometric titration step is to change the SOC of the electrode, and the effect of side reactions is neglected. The role of relatively long relaxation steps is to ensure a near-equilibrium state ($dE/dt \approx 0$) before the next titration step. Therefore, relaxation periods are terminated as soon as a pre-

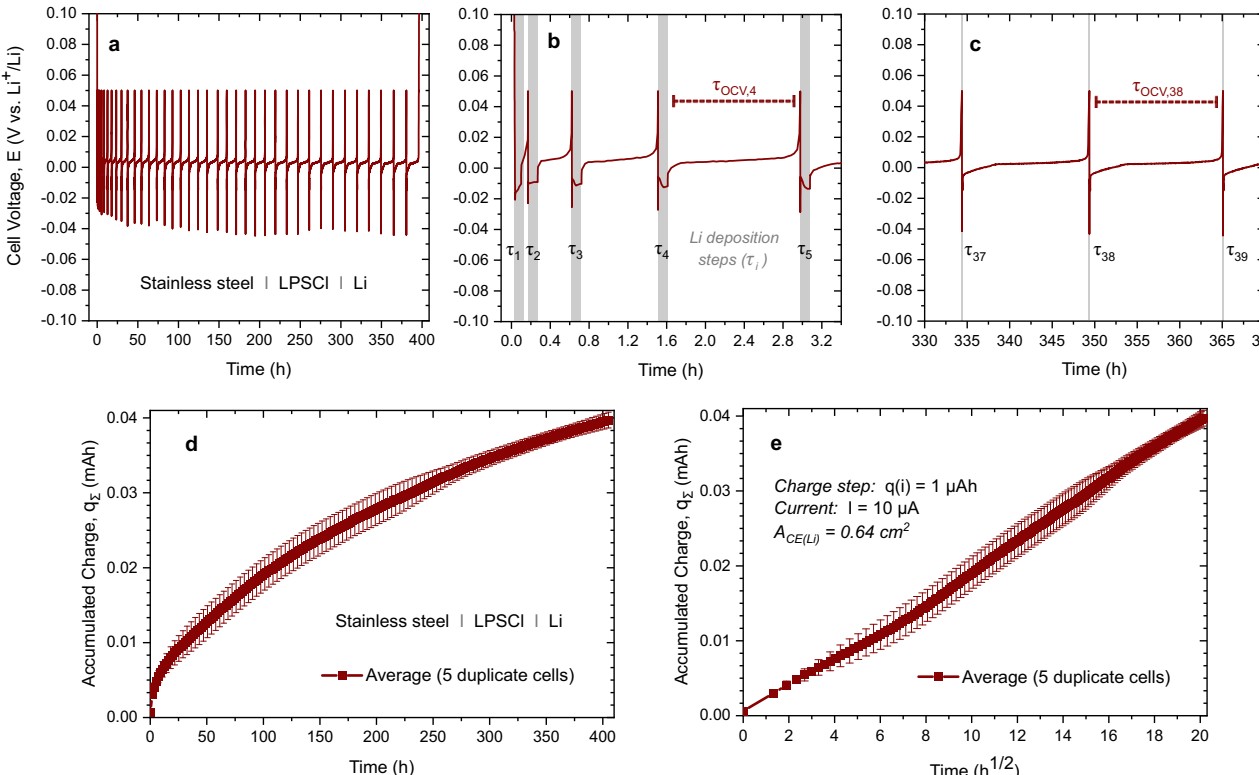

**Fig. 1 | Coulometric titration time analysis (CTTA) results. a** Results of LPSCl solid electrolyte in a stainless steel | LPSCl | Li cell configuration at $T = 25\,°C$ and $p \approx 13$ MPa. The potential profiles are shown for zoomed-in time periods from an early (**b**) and a later stage (**c**) of the experiment. The durations of constant current lithium deposition steps (i.e. lithium titration) are shown in gray. The regions between each lithium deposition steps correspond to OCV resting states ($\tau_{OCV,i}$). The sum of accumulated capacity over time is shown as function of time in (**d**) and as function of square root of time in (**e**). The error bars correspond to standard deviation. Source data are provided as a Source Data file.

defined equilibrium condition is reached. In contrast, in our CTTA technique, the aim of the titration step is not to change the 'SOC', but rather to keep it constant (applied in small steps to compete against electrolyte side reactions). Additionally, the relaxation period is not terminated after reaching the equilibrium. Instead, a secondary resting period (OCV-state) is applied. This additional step is maintained until the cell equilibrium is disrupted by the completion of side reactions (i.e. complete consumption of the previously titrated charge). This new methodology makes the quantification of electrolyte side reactions possible by accounting for the actual electrode–electrolyte interactions.

As detailed in the experimental section, the experimental setup consists of a CC (WE) which is in direct contact with the electrolyte to be tested, and a counter (and reference, RE) electrode (CE) with a constant potential and excess lithium inventory. This is a model and test case of the so-called anode-free cell concept. For our analytical technique the CE is required to have a constant potential throughout the whole experiment, in order to properly evaluate the potential of the WE. Here, we demonstrate this new technique using a stainless-steel CC as WE, $Li_6PS_5Cl$ (LPSCl) as the SE and lithium metal as CE and RE. The results of an exemplary CTTA experiment are shown in Fig. 1. The typical OCV of such a cell is 2.3–2.5 V vs. Li+/Li before the start of the experiment. We like to note that the WE potential in the pristine state is a priori thermodynamically not well defined, as the stainless steel | LPSCl interface does not represent a well-defined and reversible redox system. However, the initial voltage of 2.3–2.5 V suggests that minor sulfur redox may establish an initial and somehow fragile equilibrium.

In the first titration step ($\tau_1$), we apply a current $I(\tau_1) = 10\,\mu A$ (15.6 $\mu A\,cm^{-2}$) for 0.1 h and thus provide a charge $q(\tau_1) = 1\,\mu Ah$ (1.56 $\mu Ah$

$cm^{-2}$) to the WE, i.e. the stainless steel CC. During this step, the cell voltage drops below $E = 0$ vs. Li+/Li and lithium metal is deposited on the CC (depending on the electrolyte reactivity, some charge can already be consumed during this titration step by side reactions, instead of leading to lithium deposition). After the titration step, the first OCV period begins and the cell voltage $E(t)$ is recorded. In an ideal (limiting) case, i.e. with a stable electrolyte and no side reactions, the cell potential would relax quickly to $E = 0$ V after a certain time since lithium metal defines the potential of both WE and CE, i.e. a symmetrical cell state is established. In this ideal case, the potential would stay constant at $E = 0$ V for infinite time. If the electrolyte is not stable and side reactions occur, the 'titrated' Li metal is consumed gradually by these parasitic reactions. As long as there remains some lithium metal at the WE, the potential remains as $E = 0$ V. However, after a certain time, the Li metal will be completely consumed, the WE potential is not fixed anymore, and the voltage of the cell increases. In the case of LPSCl SE, we observed that this potential increase occurs quickly after the first titration step (see Fig. 1b). When $E$ reaches 0.05 V (i.e. our criterion chosen to end the OCV period after titration), another identical titration step is applied (such steps are shown in gray) and the cell voltage drops below $E = 0$ V again, and fresh Li metal is deposited on the CC. In the following second OCV period, we find that the time required for the complete lithium consumption is longer than the previous OCV period, showing that the rate of the side reactions is lower after the second titration step. These titration steps and subsequent OCV periods are repeated over an extended period of time and the time required to consume lithium metal in each step is monitored. As can be seen in Fig. 1c, consumption of the same amount of lithium metal takes longer and longer as the experiment proceeds—indicating passivating behavior of the side reaction products, i.e.

formation of an SEI. The amount of accumulated charge consumed in side reactions ($q_\Sigma = \Sigma\ q(\tau_i)$ = number of titrations times step charge) with respect to the duration of the experiment is shown in Fig. 1d. In this way it is possible to quantify the SEI growth precisely over time. As can be seen in Fig. 1e, the growth follows a linear dependence with respect to the square root of time. After nearly 400 h, the accumulated charge reaches almost $q_\Sigma = \Sigma\ q(\tau_i) = 40\ \mu Ah$ (i.e. $q_{A,\Sigma} \approx 60\ \mu Ah\ cm^{-2}$). Assuming homogeneous lithium metal deposition resulting in a planar and non-porous SEI film, a rough estimate of the SEI thickness (see Supplementary Table 1 for detailed information on the moles and volumes of reactants and products) can be made assuming the decomposition reaction

$$Li_6PS_5Cl + 8\,Li \rightarrow Li_3P + 5\,Li_2S + LiCl \qquad (1)$$

In the case of LPSCl, 1 $\mu Ah\ cm^{-2}$ charge (corresponding to $\approx 3.7\cdot10^{-8}\ mol\ cm^{-2}$ Li) would result in an SEI thickness $d \approx 9\ nm$ (assuming a compact mixture of $Li_2S$, LiCl and $Li_3P$, and absence of gaseous products). This would correspond to an estimated SEI thickness of $d \approx 315\ nm$ after 1 week and $d \approx 540\ nm$ after about 400 h (-17 days). This is significantly thicker than previous estimates (i.e. few nm)[53] derived from impedance measurements and estimates of the SEI conductivity, but in agreement with a more recent experimental study from our group showing the formation of a 235−305 nm thick SEI layer (after 1 week) via combined TOF-SIMS and atomic force microscopy (AFM) measurements[54]. We like to add that—according to classical solid state reaction kinetics with diffusion control—SEI formation should not come to an end at a certain time. However, it appears that finite SEI growth can be observed in some cases[55]. The reasons are yet not clear, and further work in the fine tuning of the SEI is required.

In summary, our titration technique is quite powerful in characterizing side reactions occurring between lithium metal and sulfide-based LPSCl electrolyte quantitatively, i.e., it can provide valuable insight in understanding the stability of electrolytes. It is versatile and can be easily applied to other electrodes—while also allowing to study the influence of important cell operation parameters on the SEI characteristics and cell degradation, e.g. temperature, current density, etc. These experimental opportunities, as well as potential limitations of the CTTA technique, will be discussed in the following sections.

### Effect of titration step current and charge

The choice of titration step current and charge is expected to affect trends of charge accumulation. In contrast to CV (or LSV) experiments, lithium metal is deposited on the WE in each titration step and this induces local volume changes at the interface between the CC and the existing layers of reaction products (i.e. SEI). Mechanical damage such as crack formation may occur due to stress caused by such volume changes and may affect the amount of charge consumed in side reactions. Furthermore, dendritic lithium growth into SE separators is one of the major and well-known problems for lithium metal electrodes. We expect that dendritic lithium growth is favored under test conditions with higher titration current density and titration charges. This means that lithium metal may penetrate through the existing SEI layer and thus be exposed to fresh electrolyte regions in each subsequent titration step. Two important outcomes of this phenomenon would be an increased rate of side reactions and a higher probability of dead lithium formation (i.e. lithium metal which has electronically or ionically become isolated). Both of these are important issues for metal electrode-based cells and therefore their prevention is crucial for practical battery application. It is therefore very important to characterize the effectiveness of any approach aiming to mitigate/prevent these issues. We believe that CTTA can be a valuable tool for this purpose since the deposition of higher surface area Li dendrites could decrease the $\tau_{i\text{-OCV}}$ by accelerating side reactions and increasing the likelihood of dead-lithium formation (see Fig. 2).

In Fig. 3a, the effect of current is shown for a relatively small charge step of 1 $\mu Ah$ in stainless steel | LPSCl | Li cell configuration. It is seen that both currents tested here (0.01 mA and 5 mA) result in almost identical CTTA trends showing that the higher current density (at a small charge step) is not sufficient to change the Li deposition behavior (in terms of Li | SE interface area). However, increasing the charge step to 10 $\mu Ah$ (at 5 mA) has a significant impact on the CTTA trends (see Fig. 3b). While the same amount of charge (-0.04 mAh) has accumulated in nearly 400 h with 1 $\mu Ah$ step charge, it took significantly shorter time with 10 $\mu Ah$ charge step showing the high tendency of dendritic lithium growth into LPSCl solid electrolyte. The evolution of the internal cell resistance for these two cells (as calculated from the ohmic potential drop directly after the titration step) is also shown in Supplementary Fig. 1. As seen from these results, the resistance values are in a similar range for both cells. The cell tested at 5 mA current and 10 $\mu Ah$ step charge initially has a lower resistance and it does not show a logarithmic growth behavior through the entire test duration (e.g. rather a linear-like dependence is seen after 25 h). These trends in Fig. 3b and Supplementary Fig. 1 are not surprising since high current and charge would favor deposition of lithium and SEI growth in a three-dimensional morphology towards the electrolyte by forming longer whiskers instead of a planar morphology. Therefore, the effective surface area of the electrode is higher than for planar growth. This results in a higher degree of side reactions and dead lithium formation, thereby causing additional accumulated charge in Fig. 3b (it should be noted that separating these two processes would require additional post mortem analysis). A lower cell resistance is also observed initially, however, as also observed in Supplementary Fig. 1, such cells are expected to show higher resistance eventually due to excessive extent of side reactions. It should be noted that the choice of 10 $\mu Ah$ charge steps (at 5 mA) led to short-circuiting of some cells, and therefore, smaller charge steps or alternative counter electrodes (e.g. In/InLi) are recommended for high current/charge step tests.

### Effect of temperature

The extent of electrolyte side reactions is expected to depend on the cell operation temperature. This dependence can simply be explained by the increased rate constants of side reactions which is expected to follow an Arrhenius-type behavior; however, the degree of side reactions (and also the passivation ability of the SEI) may depend on a number of additional factors such as electrode interactions (i.e. crosstalk), lithium plating characteristics, SEI solubility in liquid electrolytes, etc[56–58]. In the case of SE such as LPSCl, the dendritic lithium growth can also affect the CTTA results as discussed in the previous section and therefore contribute to the temperature dependence of side reactions (e.g. due to changes of mechanical properties of Li and LPSCl at different temperatures). It is therefore intriguing to test SEs at different temperatures using the CTTA technique. In Fig. 3c, the comparison of three different test temperatures is shown for the stainless steel | LPSCl | Li cell configuration (tested at 0.01 mA with 1 $\mu Ah$ charge step). As these results indicate, the degree of side reactions is more severe at elevated temperatures. Interestingly, the charge accumulation trend at 10 °C is slightly different as compared to higher temperatures since it starts to follow more of a linear behavior after nearly 150−200 h. More prominent trends are observed at −20 °C, as the slope of the accumulated charge as function of the square root of time starts to deviate from linearity after a certain duration of the experiment (see Supplementary Fig. 2a). The logarithm of the slopes during the initial test periods as function of $1/T$ (Supplementary Fig. 2b) shows Arrhenius-type behavior. We assume that the deviations from Arrhenius-type behavior at later stages of the experiment at low temperature indicate lithium plating in or on the SEI since dendritic lithium growth is favored at lower temperatures[59,60]. It is also important to consider potential thermal effects, e.g. local temperature variations due to Joule heating, particularly at high charge steps / high currents.

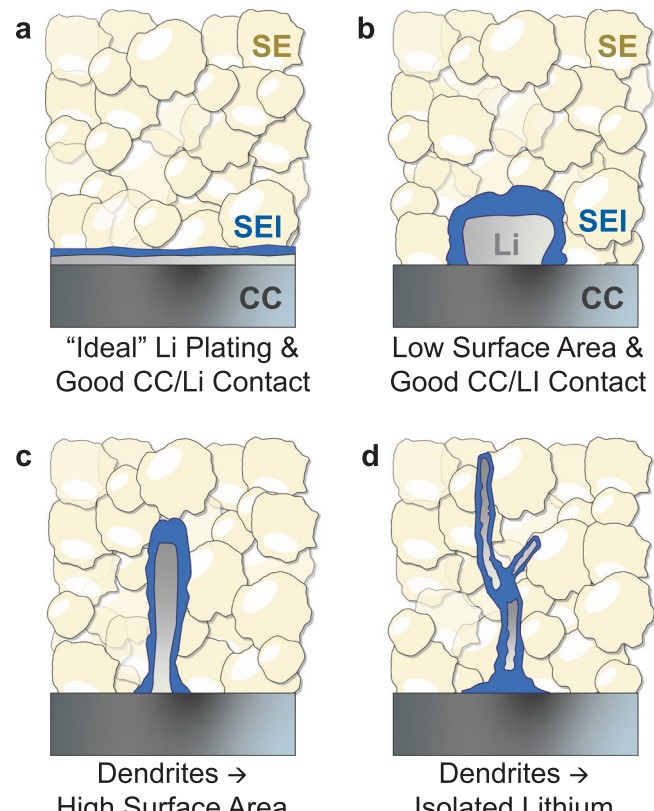

**Fig. 2 | Schematic illustration of various lithium metal deposition ($\tau_i$) morphologies. a** Ideal lithium plating morphology. **b**–**d** Other morphologies possible under different test conditions. Dendritic growth can affect the CTTA results by (**c**) an increase of active electrode surface area and (**d**) loss of electrical contact with the CC.

### Effect of current collector material

The examples of commonly used CCs for negative electrodes are thin metal foils of Cu, Ni, and stainless steel[61]. Among these materials, Cu has been the most commonly used CC in commercial Li-ion batteries despite its high Li solubility limit (up to ~22 at%) at room temperature according to the Cu-Li phase diagram[62]. The widespread use despite lithium solubility can be attributed to sluggish lithium diffusion into the Cu CC. However, some recent studies indicate that the amount of lithium diffused into the Cu (and also Ni) CCs may not be negligible as generally assumed, and thus can contribute to the capacity fading (e.g. lithium trapping) observed in full cells[63]. In the CTTA technique, the time $\tau_{OCV,i}$ required to consume Li metal deposited in each titration step $i$ is analyzed. If such a diffusional Li loss into the CC exists, it will also contribute to the test results. Therefore, we tested some commonly used CC materials in the CC | LPSCl | Li cell configuration (at 0.01 mA current with 1 µAh charge step). Aluminum is known to alloy with lithium at low potentials forming Al-Li intermetallic compounds and therefore its use as a negative electrode CC is not possible[61]. This can easily be demonstrated in Fig. 3d as the accumulated charge quickly rises to high levels in a very short time. In fact, lithium is consumed simultaneously during the coulometric titration steps rendering the experiment simply to a constant-current discharge procedure (which could be avoided if higher titration currents were chosen). In the case of Cu, CTTA trends do not follow a square root time dependence and the accumulated charge is considerably higher as compared to Ni and stainless-steel CCs. This may be caused by Li diffusion into the Cu CC, however, more severe chemical instability of LPSCl in contact with Cu (e.g. Cu corrosion by formation of Cu-S compounds)[64] also needs to be considered as this can accelerate the Li consuming

electrolyte side reactions as well. As will be shown in the next section, the contribution due to lithium diffusion into Cu is indeed rather small, and the main reasons for the lithium loss are the side reactions that occur between Cu and LPSCl. Of the materials tested here, stainless steel and Ni CCs show the lowest amount of lithium loss over time and are therefore best suited for the future CTTA experiments aimed at investigating lithium loss caused by electrolyte side reactions with sulfide solid electrolytes. We note that other choices of optimum CCs may apply in the case of other electrolyte types.

### Comparison of different electrolytes

A number of different SEs were tested in the stainless steel | SE | Li cell configuration and the results are shown in Fig. 4. Among the SEs studied here and as expected, LLZO is the most stable SE at low potentials when in direct contact with lithium metal[19], and therefore only minor side reactions are found for this electrolyte.

In the case of LLZO, the stainless steel | LLZO WE configuration can be used to identify the effect of some additional factors that can contribute to the measured accumulated charge. For instance, as discussed in the previous section, some of the lithium deposited in each titration step might diffuse into the CC and affect the results. Apart from this diffusion related loss to the CC, stainless steel (or many other metals) will have a native oxide layer on its surface and this oxide layer can be reduced in contact with lithium and therefore add a small contribution to the experimental results. In order to investigate this effect, we performed an in situ lithium deposition experiment in the XPS (technical details reported elsewhere[65]) on pristine stainless steel foil. Upon Li deposition, native oxide layers are reduced to their metallic counterparts (e.g. Fe, Cr) with the formation of $Li_2O$ (see Supplementary Fig. 3 for the in situ XPS results). Additionally, despite the careful cell assembly in the glovebox and use of an airtight cell casing, residual amounts of oxygen/moisture may still be present inside/on the cell parts and can affect the test results since the reaction with such contaminants could also cause Li loss. Lastly, even though LLZO is known to be stable towards lithium metal, it is still not clear whether LLZO is reduced to some small degree since the formation of a very thin lithiated LLZO interphase could form[66] and kinetically stabilize the SE. As seen in Fig. 4 (see also the inset), accumulated charge values of the cell with LLZO electrolyte are significantly lower than of the cells with sulfide electrolytes (in LLZO-based cells, some additional loss due to dead Li formation could also be expected due to inferior CC | SE contact as compared to sulfide SEs). The results confirm that the reduction of LLZO SE and the sum of all the other possible effects described above is relatively small (~6 µAh cm$^{-2}$ after ~200 h and ~8 µAh cm$^{-2}$ after ~400 h). Therefore, it can be concluded that the CTTA results observed for the other electrolytes mainly reflect the degree of SE side reactions and SEI growth. The accumulated charge results of LLZO-based cells can also be subtracted from the results of cells with different SEs (tested under identical conditions) in order to obtain a quantitative estimate of side reactions caused solely by the SE reduction reactions.

In the light of the results obtained from the LLZO-based cells, we suggest that CTTA testing of cells with a (relatively) stable SE can be a valuable tool to determine the lithium loss caused by lithium diffusion into the CCs. Due to practical relevance (e.g. commercial LiBs with Cu CC), such cells with Cu CC were tested in a standard press-cell casing with 20 µm-thick Cu metal foil as well as in a pouch cell with a thermally deposited 100 nm thick Cu film, since the type of foil and the cell casing might also affect the results (see Supplementary Fig. 4 for the results and further discussion). In the standard cell casing, the accumulated charge was nearly doubled as compared to a stainless steel CC. Nevertheless, the amount of charge accumulated after 400 h was still low (~14 µAh cm$^{-2}$). In the case of the pouch cell (100 nm thick Cu film), the accumulated charge after 400 h was even lower with ~9 µAh cm$^{-2}$. We estimate that—even if the measured charge in CTTA is

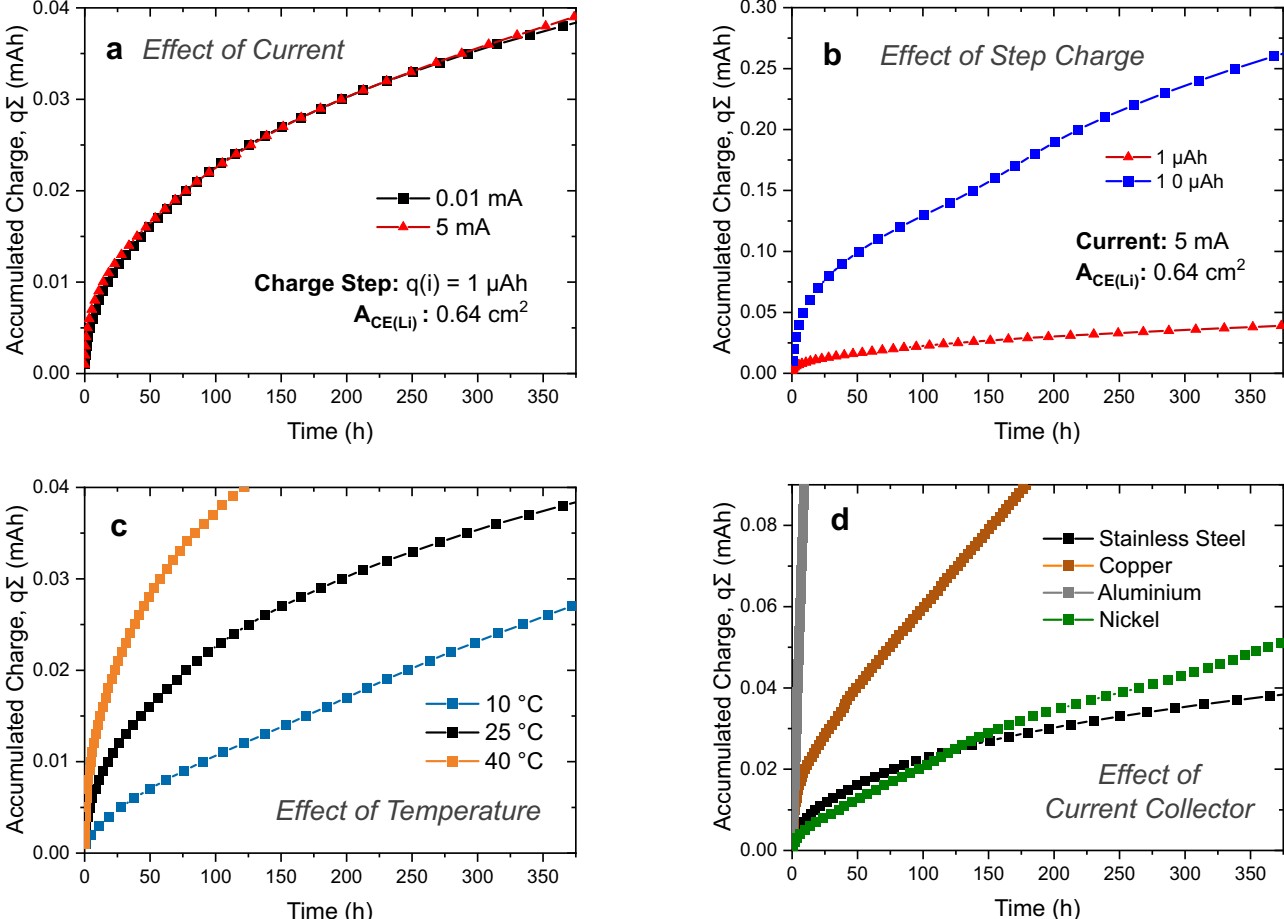

**Fig. 3 | The effect of titration parameters, temperature and current collector.** **a**, **b** CTTA results obtained from stainless steel | LPSCl | Li cells under different test conditions. In (**c**) titration experiments at different temperatures were performed at 0.01 mA with a titration charge step of 1 μAh. In (**d**), the effect of current collectors (CC) is shown in CC | LPSCl | Li cell setups tested at 0.01 mA with a titration charge step of 1 μAh. In all cells, the stack pressure of cells was $p \approx 13$ MPa. Source data are provided as a Source Data file.

assumed to be caused only by lithium diffusion loss into the CC—for the 20 μm thick Cu CC it would take at least ~7 years for a 3.5 mAh cm$^{-2}$ battery to lose 5% of its initial capacity. To the best of our knowledge, such precise considerations on the role of the CC have never been reported.

In previous reports, the resistance evolution of symmetrical cells with Li metal electrodes (as measured with EIS) showed similar behavior for LPS and LPSCl solid electrolytes which indicates a similar degree of side reactions for both electrolytes[46]. As seen in Fig. 4, we also observe similar trends in CTTA experiments for these two electrolytes. On the other hand, LGPS solid electrolyte is known to form reaction product layers lacking sufficient passivation properties due to the presence of electronically conductive reaction products (e.g. Ge or Li$_x$Ge alloys)[22]. This results in a rather fast lithium consumption over the duration of the CTTA experiment. Already after 1 week, the accumulated charge reaches 0.45 mAh corresponding to a Li loss close to ~0.7 mAh cm$^{-2}$. This would correspond to a 20% capacity loss already after just 1 week of operation for a commercial battery with an areal charge density of 3.5 mAh cm$^{-2}$.

The results presented here highlight the potential of CTTA to provide precise quantitative information for a better understanding of side reactions and interdiffusion between Li metal, CCs and SEs. This experimental set up (i.e., the anode free cell configuration) with thin stainless-steel CCs has also an advantage in terms of cell disassembly which is rather important for the post mortem analysis. In this set up, stainless steel foil and SE pellets can be separated easily from each other after the cell disassembly and can be further characterized ex situ with different analytical tools. In the following, ex situ characterization results of such samples (e.g. via SEM, XPS, ToF-SIMS) are discussed to better understand the SEI growth during a typical CTTA experiment.

## Morphological characterization of the stainless steel−LPSCl interphase

The morphological analysis of the SEI layer forming between the stainless steel CC and the LPSCl SE was first performed with scanning electron microscopy (SEM). Cell operation was stopped after a specific time and the cell was opened in an argon glovebox. The stainless-steel CC was removed from the SE pellet and both sides were investigated in SEM. As seen in Fig. 5a, b, the LPSCl side remained almost unchanged except the formation of spherically shaped particles which are found only in certain areas on the surface. At higher magnifications (see Fig. 5c), triangle and rectangle shaped features on the nano-scale are observed at some parts of the large SE particles. On the CC side, differences before and after the test are significant and spherically shaped particles are found on the sample surface (see Fig. 5e–i). Some of these particles were fractured during the removal of current collector (which was done directly after cell opening in the glovebox) indicating that these particles were formed during the experiment which was performed under 13 MPa stack pressure (see Supplementary Fig. 5). In only few areas, the initial lithium deposition sites, formed in a web-like morphology[67,68], are clearly visible together with spherical-like features (Fig. 5h). Independent of these features, as seen in Fig. 5f,

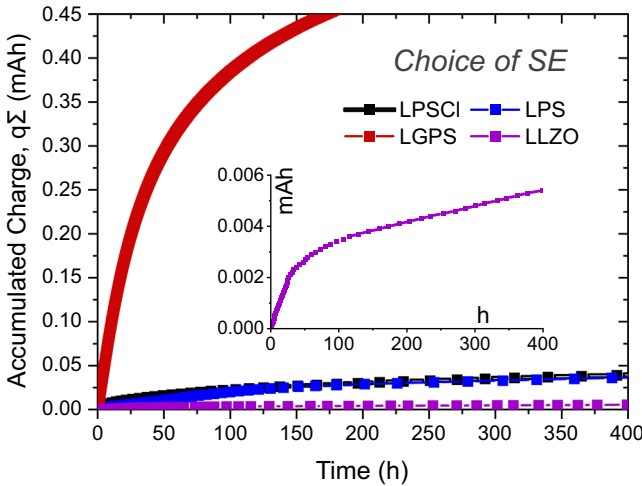

**Fig. 4 | Comparison of different solid electrolytes.** SEs in contact with stainless steel are tested at 0.01 mA current and 1 μAh charge step ($p \approx 13$ MPa) with the exception of LLZO which was tested at 0.001 mA current and 0.1 μAh step size due to comparably higher resistance of this cell. The inset shows the data for LLZO-based cell in a zoomed-in time axis view. Source data are provided as a Source Data file.

the CC is covered with a two-dimensional film that has a fine-pored microstructure consisting of smaller primary particles whose size distribution varies over the sample surface. (e.g. see also Fig. 5i). This film was also etched (via cryo-FIB) for better visualization of its thickness as shown in Fig. 5g.

Elemental analysis (EDS) was performed on the same samples. As the large area measurement results show (magnification 100x, ~1.1 × 0.8 mm²), LPSCl pellets have similar elemental composition (atomic percentage) before and after the experiment, except for an increase in oxygen content at their surface (see Fig. 5j). For the CC side, the increase of oxygen percentage is much higher and more significant, indicating reactivity of lithium and SEI products toward residual oxygen impurities that might be present inside/on the cell casing and cell components, the glovebox and the SEM transfer module/chamber. The fraction of sulfur is comparably high (particularly in comparison to phosphorous and chlorine), indicating formation of a S-rich SEI at low potentials as reported in an earlier study[54]. The small fractions of phosphorous and chlorine can be due to LPSCl particles stuck to the CC during disassembly while minor contributions from SEI products such as LiCl or $Li_3P$ are also possible.

### SEI growth on CC vs. lithium metal

In the previous section, it was shown that the SEI growth on CCs is inhomogeneous (see Supplementary Fig. 6 for additional images showing the heterogeneous regions of electrodes harvested after different test durations). There were areas where lithium nucleation/growth occurred preferentially, and inhomogeneous distribution of spherically shaped large particles (0.5–2 μm) was observed between such areas. The remaining regions were covered with relatively homogeneous planar/porous reaction layers. In the light of these observations, it becomes important to understand the local SEI growth on bare regions of the CC as compared to regions where lithium metal deposition occurs preferentially. As depicted schematically in Fig. 6a, during the titration, lithium nucleation starts at specific sites on the CC surface, e.g. at point A. It is reasonable to expect an immediate reaction between the lithium metal nucleus and the SE. On the other hand, any bare area of the CC such as point B is in direct electrical contact with lithium metal and thus subject to Fermi level alignment (i.e. metal-metal junction). As both points are also in ionic contact through the solid electrolyte, the side reactions could also start on bare CC regions such as point B. In this case, even though the cell itself is in open-circuit

state, some regions of the CC would act either as an anode (e.g. point A) or cathode (e.g. point B). Then, the electrolyte side reactions occurring at point B would result in lithium dissolution from point A. The kinetics of such reactions (or relative rate of side reactions on point A vs. B) would mainly depend on the electronic conductivity of the SEI and the ionic conductivity of the SE (assuming negligible electronic resistance between point A and B through the CC).

So far, we demonstrated the CTTA technique in an anode-free cell configuration using lithium metal as combined counter and reference electrode. A galvanic corrosion case similar to the one described above can also be created if the positive and negative ends of the cell are connected simply via an electrical cable (i.e. by external electronic short-circuiting, see Fig. 6b). If this cell is connected to a potentiostat, the amount of charge passing through the external short-circuit can be measured. For this purpose, such a cell was tested in zero resistance ammeter mode as reported previously for liquid electrolyte-based cells[69,70]. The results of such an experiment, i.e. an electrochemical noise measurement, is shown in Fig. 6c. As seen in this graph, the electric current is in the range of 1–10 μA during the first hour and gradually decreases over time indicating side reactions due to indirect contact between the CC and lithium metal, mediated ionically by the SE and electronically by the external cable. It is important to note that the rate of reactions occurring on the CC surface depends on the thickness of any existing SEI on the lithium metal counter electrode. This can be tested with a similar experiment performed on an identical cell except the pre-test storage time (the time passed between the cell assembly and the start of the cell testing). As shown in the previous sections, LPSCl SE forms a relatively thick SEI already in 1 week without causing a significant increase in the cell resistance. This proves that ionic transport through the SEI proceeds with relatively fast kinetics, thus the ionic resistance will not be rate determining in such an experiment. On the other hand, an increasing electronic resistance of the growing SEI on the lithium metal (e.g. at point A) would be the limiting factor after a certain thickness is reached. At this point, side reactions occurring on the CC would be favored, e.g. at point B. Therefore, the thicker the SEI formed on the lithium metal, the higher the measured electrochemical noise current would be in such an experiment. This is demonstrated in Fig. 6c with a second cell which was stored for 3 days (instead of 10 h) prior to the experiment.

In an alternative experiment, cells were simply short-circuited through an external cable for different time periods before the CTTA experiments were performed. In this case, during this short-circuit period, SEI growth already starts on the CC surface even though no Li metal is deposited on the CC. The growth of the SEI during short-circuit storage will result in a slower charge accumulation in the following CTTA experiment. As expected, the results of such cells (see Fig. 6d) confirm this assumption. For further examination, a similarly short-circuited cell (for 400 h at RT) was opened in a glovebox and samples were prepared for post mortem characterization. The analysis of these samples shows that the planar-like SEI growth and formation of spherically shaped particles still occur on the CC due to reductive side reactions (see also the cryo-FIB etched region in Fig. 6e). This layer is similarly S-rich, and there are oxygen-rich spherically shaped particles which are randomly distributed all over the electrode while oxygen-rich regions are similarly close to the CC (see Fig. 6f–h).

We conclude from these experiments that the consumption of lithium metal is initially caused by a direct SEI growth on the freshly nucleated Li metal itself, but also on the bare CC parts if the aforementioned alternative reaction pathways become possible after the initial SEI growth on the Li metal. These experiments also highlight the importance of the choice of the counter electrode in conventional CV (or LSV) experiments. One drawback of such experiments is that the degree of side reactions is not representative for the actual active material | electrolyte interface reactivity. However, it is shown that the choice of the CE can also induce errors in such measurements. This is

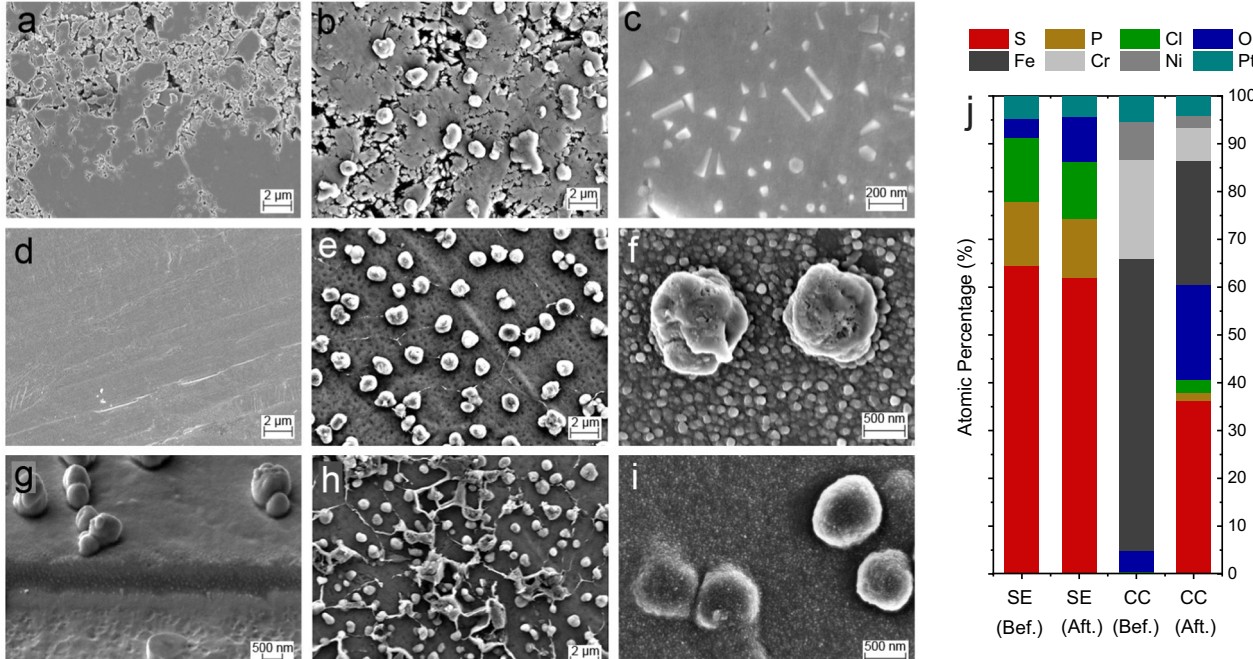

**Fig. 5 | Post mortem characterization with SEM and EDS.** SEM images of LPSCl solid electrolyte (SE) before (**a**) and after CTTA experiments (**b**, **c**). The samples were harvested from the cells after nearly 400 h of test. SEM images of stainless-steel current collectors (CC) before (**d**) and after CTTA experiments (**e**–**i**). **g** The focused ion beam (FIB) cross section of the planar SEI layers (on CC-side) prepared under cryo conditions. **h** A representative image from the area (on CC-side) where Li metal deposition is significant/visible. **j** EDS results (before and after the CTTA experiment) of LPSCl and CC samples. Source data are provided as a Source Data file.

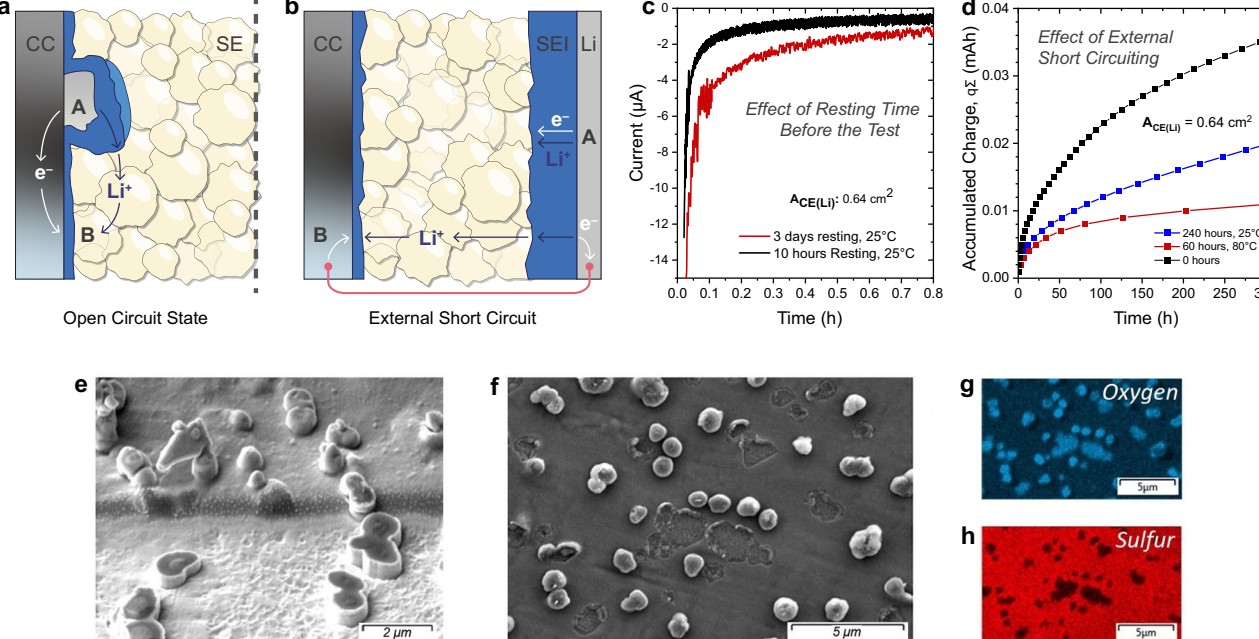

**Fig. 6 | Short circuit experiments. a** A simple schematic of the CC that is in contact with the SE. **b** A simple schematic of the anode-free cell configuration with lithium metal as the counter electrode. **c** Electrochemical noise measurements performed on anode-free cells which were subjected to two different cell storage times prior to the experiment (i.e. cells having different SEI thickness on the lithium metal counter electrode). **d** CTTA results of cells which were subjected to short-circuiting through an external cable for different durations/temperature prior to experiments ($p \approx 13$ MPa). **e** The focused ion beam (FIB) cross section image of the planar SEI layers (on CC-side) prepared under cryo conditions. The sample was harvested from a cell which was only stored (400 h, RT) under short-circuit conditions (electrodes were connected through an external cable). **f** The SEM image of the CC surface from the same cell in (**e**). During the removal of CC from the SE-side, some parts of the SEI layer were detached from the CC-side making it possible to get more information on the layered SEI structure. As can be seen in EDS maps for oxygen (**g**) and sulfur (**h**) planar regions are S-rich and spherical regions are O-rich. A second planar region which is O-rich is also buried under the S-rich layer. Source data are provided as a Source Data file.

particularly valid for determining the degree of side reactions at low potentials when the counter/reference electrode is also subject to side reactions. This is due to the fact that the experimental set up is a galvanic cell and the current being measured will also depend on the thickness of SEI already formed on the CE (or the time passed after the first contact of the CE with the electrolyte). Lithium metal is the most commonly used CE in such studies, therefore, it is recommended to use counter electrodes operating at higher potentials (e.g. partially lithiated LFP electrodes) or to add relatively long OCV steps before starting the CV/LSV tests so that the comparability between different cells is ensured.

The experiments shown in Fig. 6c, d also provide valuable information to better understand the stability of lithium dendrites growing into SE separators. If dendrites grow on CCs, which have large areas deficient of lithium metal, the dissolution rate of the lithium dendrites will be higher and thus the observation of short circuits will be delayed. On the other hand, if dendrites are formed on a planar lithium metal electrode, then the SEI growth around the lithium dendrites will not only consume Li metal from the dendrites itself, but also from the lithium metal electrode to which the dendrites are electronically connected. This would have a negative impact on the Li dendrite self-healing by side reactions since the dendrite dissolution rate would be relatively low. This effect is demonstrated by an extended testing of a CTTA cell which failed due to short-circuit caused by dendritic lithium growth (see Supplementary Fig. 7 for the test results and further discussion of this effect).

The morphological analysis of CC–LPSCl interphase by electron microscopy revealed a need for further investigation of side reactions occurring on distinct regions of the CC. In this section, we show that consumption of lithium metal is not solely a result of SEI growth on the freshly nucleated Li metal surface, and morphologically diverse SEI growth on bare CC regions is possible via local galvanic corrosion pathways. In the next section, we investigate this apparently complex interphase formation further through the use of XPS and ToF-SIMS techniques.

## Surface analysis of the current collector–LPSCl interphase

Parts of the electrodes harvested after the CTTA experiment have been analyzed with XPS and ToF-SIMS, as these techniques can provide further information about the chemical environment and composition of elements with a much higher surface sensitivity than SEM. The XPS results of the S 2$p$, Cl 2$p$, O 1$s$ and Li 1$s$ spectra are shown in Fig. 7. In the S 2$p$ spectra, main differences are observed for the CC side of the electrode in which the Li$_2$S related peaks have the largest contribution. In the case of the SE side, relative intensities of different species are similar except for a slight increase in the P-S-P set of peaks. For the Cl 2$p$, peak positions remain unchanged for both CC and SE sides. On the CC side, presence of LiCl is possible as this compound has a similar Cl 2$p$ binding energy as the LPSCl electrolyte. Interestingly, this sample does not show any significant peaks in the P 2$p$ spectra (see Supplementary Fig. 8) confirming the phosphorous deficiency of SEI films.

As seen in Fig. 7e, significant changes in the relative S/Cl/P ratios are observed particularly at the CC side. These results support the formation of SEI films with an inhomogeneous/layered structure which is S-rich and P-deficient near the CC. Atomic fraction trends also show a gradual increase in oxygen content after the titration experiment which is more significant for the CC side. These trends become more evident after performing some surface sputtering (see also Supplementary Fig. 9), indicating that oxygen is a component of the SEI rather than being merely a part of surface contamination layers. The main contribution to the oxygen signal comes from the main peak at around 531.5 eV where compounds such as LiOH and Li$_2$CO$_3$ are expected, however, the trends of the C 1$s$ spectra (see Supplementary Fig. 8) show that contributions from carbonate species probably have a rather low impact. The appearance of a new peak at 530.0–530.4 eV corresponds to metal oxides while a third peak at around 533 eV is only

observed at the LPSCl side which can be ascribed to oxygenated phosphorus. The binding energy for the metal oxide peak is in close proximity with the transition metal oxides, however, no intensities could be detected in the Fe 2$p$ and Cr 2$p$ spectra (even after surface cleaning via Ar$^+$ sputtering). This indicates that the metal oxide peak may be assigned to lithium oxides. The binding energy of Li$_2$O is expected at ~528.5 eV[71,72], however, it could be observed at higher binding energies due to differential charging effects[73], if it is buried deep in the SEI. After subsequent sputtering steps (see Supplementary Fig. 9), oxygen levels reach a saturation point and a new peak at ~528.5 eV corresponding to Li$_2$O emerges. The formation of Li$_2$O at low potentials (e.g. 0 V vs. Li$^+$/Li) is expected (see also Supplementary Fig. 3 for in situ XPS experiment results of Li deposition on stainless-steel CC) and has been shown via operando XPS and HAXPES experiments performed with sulfide solid electrolytes[74,75]. Nevertheless, the formation of Li$_2$O as a result of Ar$^+$ sputtering is possible[76,77] and therefore a quantitative analysis is not possible.

In the case of Li 1$s$ spectra, the main difference is observed for the CC-side as a second peak located at ~54.2 eV emerges following the titration experiment. This energy region corresponds to possible SEI components such as Li$_2$S and Li$_2$O, however, multiple peak fitting is avoided due to overlapping binding energies of such compounds. The analysis of samples tested for shorter durations (Supplementary Fig. 10) indicates a thinner SEI layer with a reduced coverage of the CC as could be expected from the SEM observations. For the externally short-circuited sample (400 h), the trends are similar to a CTTA sample that was tested for the same duration (see Supplementary Fig. 11).

Further complementary characterization was conducted with ToF-SIMS since it offers chemical information from the SEI layers with high lateral and depth resolution and a rather high sensitivity. Similar to the XPS analysis, both the CC side (stainless steel) and LPSCl side of the working electrode were analyzed, and the results are shown in Fig. 8. Depth profiling performed in spectrometry mode confirms that the SEI film is mainly stuck to the CC side and has a layered structure (see Fig. 8a). The representative ion signal for the CC (FeO$^-$) shows a two-step increase in intensity during depth-profiling. The second increase is similar in intensity difference (note the logarithmic scale of intensity) and occurs at a similar ion fluence where considerable intensity drops are seen for the LiS$^-$ intensities. Therefore, the region up to this point (where the FeO$^-$ signal is at its maximum) can be approximated as the boundary for the SEI region. The decrease of signal intensities with further sputtering is likely a result of changes in ionization probabilities and matrix effects upon a complete removal of SEI species from the CC. In contrast to S$^-$ and LiS$^-$ signals, Cl$^-$ and LiCl$^-$ intensities decrease at an earlier stage of sputtering. In the case of oxygen, different oxygen-containing ions show individual profiles. For instance, the OH$^-$ ion intensity remains nearly constant at the beginning and later starts to decline while O$^-$ and LiO$^-$ signals gradually increase in the same region. The maxima of O$^-$ and LiO$^-$ ion intensities are also observed in a region deeper from the surface, indicating a layered microstructure of the SEI film also with respect to oxygen species. For the LPSCl-pellet side (see Fig. 8b), PS$_3^-$ signals can be used to represent unreacted LPSCl regions. Here, intensity changes in PS$_3^-$ or SEI related peaks (e.g. LiCl$^-$, LiS$^-$) are mainly observed during the early stages of sputtering which supports that the SEI regions were mainly stuck to the CC during the sample preparation.

The depth profiling measurements were also performed for additional samples obtained from CTTA experiments after different time periods (see Supplementary Fig. 11). After 2 and 10 µAh charge accumulation, inhomogeneity of the sample surface was observed as some regions showed a profile similar to the pristine stainless-steel CC. This is not surprising as such local inhomogeneities were also observed in SEM analysis of these samples (see Supplementary Fig. 6). In the case of a ~400 h short-circuited cell, the general trends were similar to the CTTA sample tested for the same duration, i.e. ~400 h.

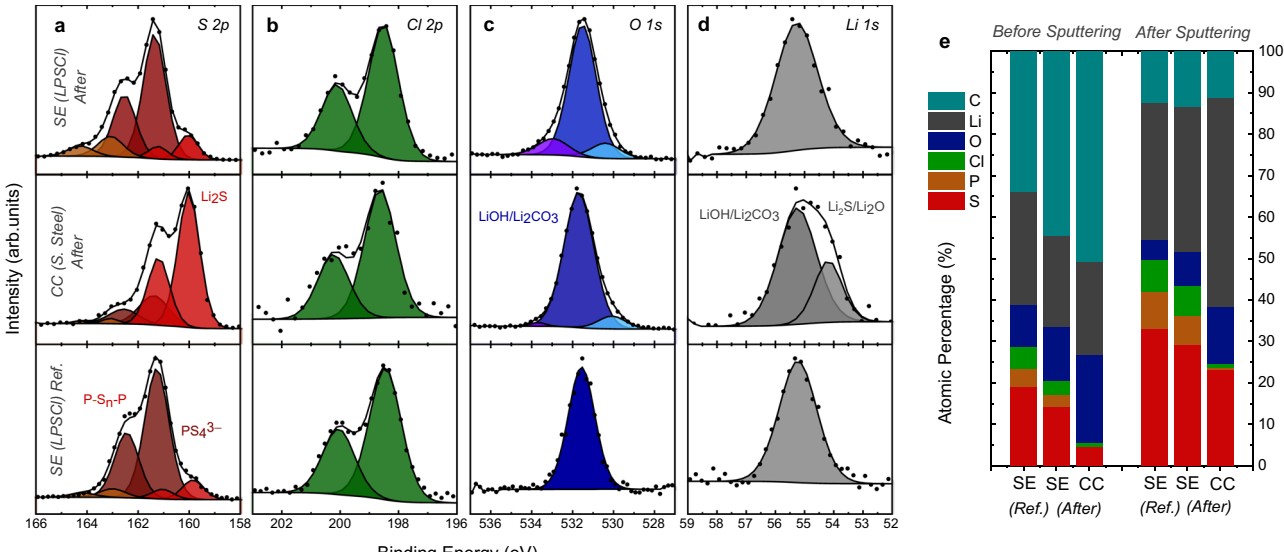

**Fig. 7 | Post mortem characterization with XPS.** Normalized (**a**) S 2*p*, (**b**) Cl 2*p*, (**c**) O 1*s* and (**d**) Li 1*s* spectra of reference LPSCl electrolyte, stainless steel CC after the CTTA experiment and the LPSCl pellet side facing this CC. Elemental concentrations are shown in (**e**) before and after gentle Ar⁺ sputtering (4 min at 0.5 kV). The spectra in (**a**–**d**) were measured before sputtering. Source data are provided as a Source Data file.

Further characterization of SEI films formed on the CC was performed by preparing wedge-shaped craters via sputtering of a rectangle by increasing the sputter dose density gradually from left to right side. In this way, it is possible to visualize the depth-dependent structure as well as the lateral structure of the SEI[54]. A smaller section of such a crater (from the middle region) is shown in Fig. 8c as an overlay of S⁻ and O⁻ ion signals and in Fig. 8d for OH⁻ ion signals. The spherically shaped particles which are still visible on the CC substrate seem to have a core-shell structure with S-rich layers surrounding an O-rich core. Interestingly, O-rich regions also seem to have a heterogenous structure as higher OH⁻ signal intensities are collected from the outer regions. For a better visualization of the layered SEI structure, a wedge-crater was prepared on a sample tested in a longer CTTA experiment (nearly 1200 h). In Fig. 8e, a region from the wedge-shaped crater is shown as an overlay image of S⁻ and O⁻ ion signals. Also, in Fig. 8f, the beginning of another wedge-crater (with a shallow depth) is seen next to the unsputtered region which is visible on the left side (Cl⁻, S⁻ and O⁻ ion intensities are overlayed with their respective colors).

The ex situ characterization of samples obtained from the CTTA experiments shows that the SEI film growing at the CC | LPSCl interface is rather thick and it can be observed even with SEM/EDS and FIB/SEM. The Li deposition and the resulting SEI growth on the CC are spatially not homogeneous and the SEI develops a layered structure with regions rich in different elements and compounds. From our previous work, the SEI formed at the Li | LPSCl interface region is known to be layered with $Li_2S$-rich regions close to the Li-side, and Cl-rich/P-rich regions close to the LPSCl-side[54]. In the present work, lithium metal is deposited on a stainless steel CC in an anode-free cell configuration which is a practically relevant experimental arrangement (e.g. due to presence of contaminants in/on cell components, in situ Li plating in a confined space, fresh Li metal exposure in each titration cycle, etc.). Apparently, this alters the SEI microstructure as we found regions where large spherically shaped particles nucleated within rather planar films. The spherically shaped particles have a core-shell structure with an oxygen-rich core surrounded by a sulfur-rich shell. Similarly, the planar regions possess an oxygen-rich layer buried under a mainly S-rich film (close to the CC). A combined XPS and ToF-SIMS analysis suggests that the S-rich and O-rich regions consist of $Li_2S$, and $Li_2O$/LiOH, respectively. Even though Cl-rich regions are found (close to the LPSCl-side), we observed no clear trends for P-rich regions. We

speculate that the reactive SEI products such as $Li_3P$ further react with cell contaminants and result in the formation of volatile/gaseous species such as $PH_3$ requiring further experimental and computational investigations in the future studies to elucidate this observation.

In this study, coulometric titration time analysis (CTTA) is presented as a new electrochemical method to investigate the side reactions occurring between metal anodes and solid electrolytes quantitatively. The technique is demonstrated using a stainless steel | LPSCl | Li cell configuration (i.e. a so-called anode-free cell) to quantify the side reactions occurring upon lithium metal plating on the CC. The technique differs from established techniques (e.g. CV, LSV, staircase voltammetry, etc.) as it does not rely on the measurement of parasitic currents by a potentiostat. Instead, lithium metal is plated on the WE (i.e. the stainless-steel CC) in rather short titration steps, and the side reactions consuming the plated lithium metal are quantified indirectly by the time analysis of the cell OCV. The main advantage of this technique is that it is possible to quantify side reactions by accounting for the actual electrode–electrolyte interactions. If the experimental parameters during the titration step are chosen such that the lithium deposition morphology is affected, it is also possible to account for morphological effects such as dendritic lithium growth through/into the existing SEI and electrolyte, making the technique quite valuable, particularly in the research for metal anode batteries. As demonstrated in this study, CTTA can be used to explore a wide range of different electrolytes and CCs as a function of all relevant parameters. It is well-suited for post mortem analysis, to correlate quantitative electrochemical information with the results obtained from other advanced analytical techniques. It only relies on the basic concept of electrochemical (coulometric) titration, i.e. the very precise deposition of a small amount of an element (in the present case a metal), and the measurement of electrode potentials as function of time−following the consumption of the titrated element by degradation reactions. As example we studied SEI formation of $Li_6PS_5Cl$ (at various temperatures) and confirmed square root of time kinetics of its thickness increase, leading to consumption of 60 µAh cm⁻² lithium (corresponding roughly to about $d \approx 540$ nm SEI thickness) in 400 h at 25 °C and in 120 h at 40 °C. Assuming that this kinetics lasts also for longer periods of time, which is a reasonable assumption, then we can extrapolate the lithium inventory loss after 1 year of contact with lithium to about ≈0.3 mAh cm⁻² at 25 °C and ≈0.5 mAh cm⁻² at 40 °C.

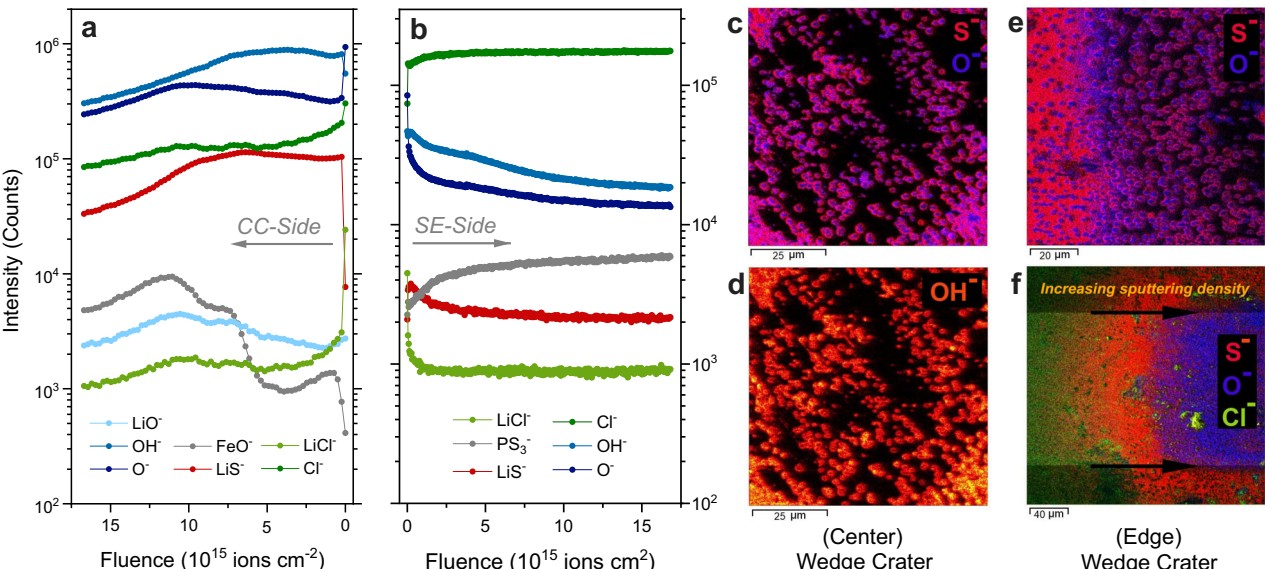

**Fig. 8 | Post mortem characterization with ToF-SIMS. a** Depth profiling results of the current collector (CC) and **b** solid electrolyte (SE) side of the WE after ~400 h CTTA experiment (i.e. 40 μAh charge accumulation). The x-axis scale in (**a**) is reversed for an easier evaluation of results. ToF-SIMS images of current collector (**c**, **d**) after wedge crater preparation using GCIB (these images were acquired near the center of the wedge crater). The beginning of the wedge for a different cell tested for longer durations (1200 h) are shown in (**e**, **f**) to demonstrate the layered structure of the SEI. Note: The wedge crater preparation was performed by gradually increasing the sputtering density from the left edge to the right edge of the crater area resulting with a horizontally increasing crater depth. Source data are provided as a Source Data file.

This shows that SEI formation and the corresponding loss of active lithium will not be found to be critical in short time lab studies but can be a severe issue for the "anode free" cell concept utilizing SEI forming sulfide electrolytes. In future studies, we will investigate chemically modified solid electrolytes, to compare the effect of SEI formation relative to LPSCl as reference material.

We believe that this broadly applicable method can become a standard procedure in battery research not only for the Li metal | sulfide SE, but also for other cell chemistries (e.g. polymer, liquid, or hybrid electrolyte cells, sodium metal anode cells, etc.) which are subject to similar issues and electrolyte degradation at the electrode surfaces.

## Methods

### Materials and electrolyte

The CC discs (9.6 mm diameter) were punched from 20 μm thick stainless-steel foils (Goodfellow, AISI 304). Those discs were ultrasonic cleaned for 3 min in isopropanol and then vacuum-dried for 10 h at 100 °C. $Li_6PS_5Cl$ (LPSCl), $\beta$-$Li_3PS_4$ (LPS) and $Li_{10}GeP_2S_{12}$ (LGPS) SE powders were purchased from NEI Corporation. For the hybrid LLZO-LPSCl cells, submicron-sized (D50: 400-600 nm) $Li_{6.4}La_3Zr_{1.4}Ta_{0.6}O_{12}$ powders from MSE Supplies were used. For the LLZO pellet-type pouch cells, $Li_{6.25}Al_{0.25}La_3Zr_2O_{12}$ SE pellets were prepared via a solid state synthesis approach as described in an earlier study[78]. In the case of LLZO pellets, one side of the pellet was coated with a 100 nm film of Cu using thermal deposition. As counter electrode, lithium discs with 9 mm diameter (6 mm for the LLZO pellet-type cells) and 100 μm thickness were used. For the In/InLi electrode, a 9 mm (diameter) indium disc (100 μm thick) was attached to a smaller lithium disc (with 4 mm diameter and 200 μm thickness) positioned at the center.

### Coulometric Titration Time Analysis (CTTA)

In this study, data were obtained from electrochemical cells with a stainless-steel disc used as working electrode (WE) and Li metal (or In/InLi composite) as counter/reference electrode (CE/RE). First, stainless steel foils of 20 μm thickness were punched into 9.6 mm diameter discs and were inserted into the press-cell setup. 90 mg of SE powder was loaded on top of the stainless steel disc and uniaxially pressed at ~400 MPa pressure at room temperature for 2 min. Later, lithium discs were inserted on top of the SE pellet and the cells were sealed. Cell assembly was performed in an argon glovebox (MBraun, $p(O_2)/p < 0.1$ ppm and $p(H_2O)/p < 1$ ppm). Unless specified, a uniaxial pressure of ~13 MPa was applied to the cells during the CTTA experiment, which was carried out at 25 °C in a temperature controlled climate chamber. The typical cell voltage was 2.3–2.5 V vs. Li+/Li before the experiment. In each titration step, a specific current was applied for a short time period, and thus, a small amount of lithium metal was deposited on the stainless-steel CC (see Fig. 9 for a schematic visualization of this process). After the titration step, the cell voltage relaxes to an open circuit voltage (OCV) of around $E = 0$ V since lithium metal is now present on both electrodes. In the next step, the cell is kept in the OCV state until the cell voltage $E$ starts to deviate from 0 V (i.e. increase to more positive values). Up to this point, side reactions consume lithium metal but the cell voltage remains constant at 0 V as long as there is still some lithium metal present on the WE CC. Deviation from 0 V indicates that the deposited lithium metal at the WE CC has been completely consumed in side reactions (or that the electrical contact with the CC is lost). We defined $E = 0.05$ V as the upper voltage limit to end this OCV period. Following this OCV period (which eventually resulted in increasing cell voltage), another titration step was applied, and the previous procedure was repeated as many times as necessary (e.g. ~400 h in this study). Different currents and titration steps (i.e. the amount of charge passed through the cell in each titration step) were applied. Detailed information is given in the main text. Typical currents were of the order of 10 μA–5 mA (15.6 μA cm⁻²–7.8 mA cm⁻²), and typical titration charges were of the order of 1–10 μAh (1.56–15.6 μAh cm⁻²).

### Ex situ characterization of the working electrode

The cycled cells were opened in an argon glovebox, and the electrode stacks were removed from the cell casing using a hand-press. The thickness and mechanical properties of the thin stainless-steel disc allowed easy removal from the SE pellet. Ex situ characterization was performed on both stainless-steel discs and SE pellets.

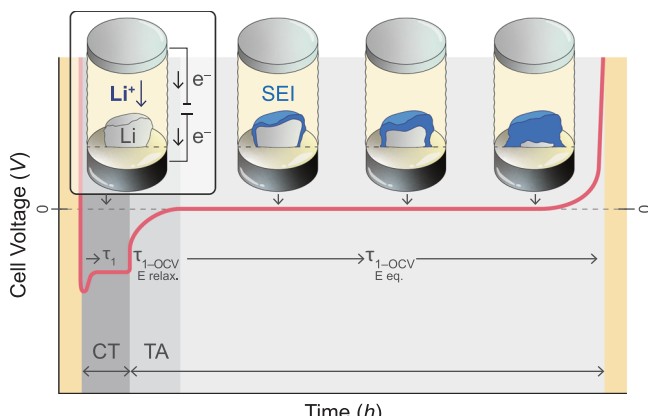

**Fig. 9 | Coulometric titration time analysis.** A schematic visualization of lithium coulometric titration on the current collector surface (CT region) and its gradual consumption (i.e. galvanic Li corrosion) by parasitic side reactions during the following OCV-period (time analysis, i.e., TA region). The forming SEI is indicated by the blue colored phase, consuming freshly titrated lithium.

**Time-of-flight secondary ion mass spectrometry (ToF-SIMS).** First, sets of samples were attached onto an Al sample holder using electronically non-conductive adhesive tapes so that the samples were not grounded. These sample holders were transferred to an IONTOF M6 hybrid secondary-ion mass spectrometer (SIMS) equipped with a 30 kV Bi cluster primary-ion gun for analysis. Samples were transferred from the glovebox with an air-tight Leica VCT500 transfer module. Depth profiles were measured in spectrometry mode (i.e. bunched mode) using a dual source column ($Cs^+$ ions, 2 kV) or gas cluster ion beam ($Ar_{1500}^+$ cluster ions, 10 kV). Measurements were carried out in the negative ion mode and a flood gun was used for charge neutralization. For the stainless-steel side of the electrodes, wedge craters with varying dimensions were prepared using a gas cluster ion beam (GCIB) with a maximum dwell time of 1 ms (applied in multiple steps until the stainless-steel became visible at the deeper side of the wedges). Image acquisition was made with the delayed extraction mode (analyzer) and imaging mode (primary-ion gun). Multiple spots were measured on each sample to ensure the reproducibility of the results.

**Scanning electron microscopy (SEM).** The same samples were coated with 4 nm Pt and then transferred to a scanning electron microscope (SEM) equipped with a field emission gun (Merlin, Carl Zeiss) using a Leica VCT500 air-tight transfer module. The operation voltage was 3 kV, and the beam current was 200 pA. With the same instrument, elemental analysis was also performed using an energy dispersive X-ray spectroscopy (EDS) detector. Additionally, the same samples were transferred similarly to a FIB/SEM instrument (XEIA3 Triglav, Xe-Plasma FIB, Tescan Orsay Holding). For minimal damage of the SEI film, a low current (0.25 nA) was used for FIB etching (at 90° angle) and the sample was cooled to −130 °C.

**X-ray photoelectron spectroscopy.** A second set of samples was prepared in a similar way for X-ray photoelectron spectroscopy analysis (XPS) and then transferred without air exposure to a PHI5000 Versa Probe II system (Physical Electronics GmbH). Measurements were performed using monochromated Al-$K_\alpha$ radiation (1487.6 eV, 200 μm beam diameter and 50 W power) and a dual beam charge compensation was applied. In detailed spectra measurements, the pass energy was 29.35 eV, the step size was 0.25 eV and step time was 50 ms. For depth profiling, $Ar^+$ ions with an accelerating voltage of 0.5 kV were used. An in situ experiment (i.e. in situ lithium deposition)[65] was also performed on stainless-steel discs which were used as CC in CTTA

experiments. For this experiment, a stainless-steel disc was placed on the sample holder using an electronically nonconductive adhesive tape. Lithium metal (target material) was mechanically scraped to remove the surface contamination, and then attached to the L-shaped sample holder which was fixed next to the stainless-steel sample. The lithium metal was sputtered by an unscanned argon-ion beam (2 kV acceleration voltage, 2.5 μA current). For the data analysis, the software Casa XPS was used, and a linear energy calibration was performed with respect to the hydrocarbon peak positioned at $E_B(C\ 1s) = 284.8$ eV. Unless specified, Shirley background subtraction and intensity normalization (divided by the maximum) was applied. Unless specified, a Gaussian/Lorentzian peak shape GL (30) was used in the peak fittings.

## Data availability
Source data are provided with this paper.

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

## Acknowledgements

This work has received funding from the Bundesministerium für Bildung und Forschung (BMBF) within the FestBatt—Cluster of Competence for Solid-State Batteries (FB2-Char, 03XP0433D). The authors thank Elisa Monte for the preparation of the schematic illustrations.

## Author contributions

B.A. conceived the idea of the CTTA technique. B.A. and J.J. designed the research project. B.A. performed all electrochemical experiments /analysis (except LLZO pellet-type pouch cell assembly), SEM/EDS and cryo-FIB/SEM measurements /analysis. L.M.R. and B.A. performed the XPS measurements/ analysis. S.K.O, B.A. and A.H. performed the ToF-SIMS measurements and analysis. T.F. prepared LLZO pellets and built LLZO pellet-type pouch cells. B.A. wrote the whole manuscript. All authors read the manuscript, provided feedback and contributed to it with useful discussions.

## Funding

## Competing interests

The authors declare no competing interests.
