## [Peer Review File · Nature Communications]

Reviewers' Comments:

Reviewer #1:

Remarks to the Author:

Aktekin et al. reported a new electrochemical method, called "Coulometric Titration Time Analysis (CTTA)" to quantify the SEI formation at the interface of Li metal and sulfide electrolytes. This method not only performs a rarely reported technique in quantifying SEI formation of side reactions by calculating the accumulated charge, but also shows the potential to analyze the issue of lithium dendrites. CTTA can also be used to explore a wide range of different electrolytes and current collectors as a function of all relevant parameters. The reviewer has acknowledged the high novelty and potential generalizability of this study and suggests publishing after minor revisions. Specific comments are outlined below:

1. According to several important review papers involving the interface characterization on sulfide-based all-solid-state batteries (e.g., doi.org/10.1007/s41918-022-00176-0; doi.org/10.1002/sstr.202100146; DOI: 10.1039/D1EE00551K), the interface issue on Li metal/sulfide include two basic aspects, namely interfacial side reaction and Li dendrites. The authors quantified the SEI formation by calculating the accumulated charge, but how can this reflect the specific contribution from side reactions or Li dendrites, respectively? Or must be combined with other characterization techniques to give this insight?
2. Consider limitations with severe interfacial side reactions: If severe chemical reactions occur between Li metal and certain SEs (e.g., metal-containing halide SEs), the kinetics of the interface reaction may not stabilize. Address this limitation of the CTTA method and its applicability under such conditions in the main text.
3. Discuss whether the CTTA method needs to account for joule heat caused by the resistance of the test cell, as it could potentially influence the calculation of the accumulated charge.
4. There is a large length on the interface characterization for the Li/LPSC, which might be deviated from the key point of this paper. Also, a lot of similar work have been published to explain the interface between Li metal and LPSC. Therefore, the reviewer suggests to re-arrange this part to make the paper more readable.
5. Attend to non-scientific issues: Ensure consistency in reference style, provide missing page/article numbers for references (e.g., refs. 65, 69, 70, 74), and maintain uniform abbreviation of journal names. Additionally, indicate the value of stack pressure in all relevant figures.

Reviewer #2:

Remarks to the Author:

This paper uses a novel electrochemical method to quantify the amount of side reactions of a solid-state electrolyte, especially LPSCl. It does an in-depth investigation within the field of electrochemistry and aims to measure the quantity of actual side reactions in solid-state electrolytes by systematically eliminating various other possibilities in their electrochemical analysis. Although the length of this paper may be not relatively concise to be published in Nature Communications and focuses much on the electrochemical analysis which might be not the best way to appeal to a broad audience, I believe that it is still considered sufficient for publication in this journal if they address these minor comments.

1. The authors claimed to quantitatively determine the amount of side reactions in solid-state electrolytes by counting accumulated charge. However, a precise explanation is needed for how the absolute quantity of charge consumed for other reasons can be excluded. The authors already mentioned the possibility of the charge consumed such as during the initial step when dropping from 2.5V to 0V and the charge consumed due to the native oxide layer in the CC. However, there seems to be an insufficient explanation of how to subtract these quantities to calculate the charge consumed during actual side reactions in solid-state electrolytes only.
2. Figures 2d, 2e, and others seem to present the results of quantifying the side reactions in solid-state electrolytes using the claimed CTTA method. However, the absolute amount to which the solid-state electrolyte has decomposed is not thoroughly discussed. It would be essential to provide more detailed information, such as the moles or thickness of the solid-state electrolyte that underwent side reactions, through figures or tables as this would be a key result of this analysis.

3. After plating/undergoing side reactions under the applied pressure conditions and the cell disassembled for SEM analysis of LPSCI or CC surfaces, how do the SEM images have spherical deposit morphologies? How does it maintain a spherical shape rather than a distorted shape?
4. Regarding the claim of P-deficient SEI formation, what is the own evidence and reasoning presented in this paper, apart from the previous literature's opinion? Is it merely because LPSCI contains 5 S atoms per 1 P atom, or are there other reasons for this claim?
5. Is there any method available to quantify oxidative side reactions of solid-state electrolytes as well as reduction reactions?

Reviewer #3:

Remarks to the Author:

The manuscript NCOMMS-23-32284 by Prof. Janek et al. discusses new method on quantifying SEI on lithium metal anode in solid state batteries. The manuscript investigates the influence of number of parameters such current density, plated capacity, temperature, and composition of the substrate on the amount of SEI formed on lithium metal with the proposed titration method. The work includes interesting results and discussion, so I think it fits to Nature Communications, however there are some comments below to be addressed:

- It is not completely clear how the "accumulated charge" was calculated in this work. Please provide more details in the beginning of the results section to clarify it.
- The initial OCV before starting the experiment was 2.3-2.5 V when stainless-steel was use which is assigned to minor sulfur redox. Could the author add information on the initial OCV with the other tested current collectors? How could the initial OCV be explained with the other current collectors (i.e. Al, Cu, Ni)? Any comments on the influence of electrolyte impurities on the initial OCV?
- The assumption of formation of SEI with thickness of almost 315 nm is based on that all the side products contribute to formation of SEI. However, another important mechanism is formation of gaseous by side reaction. Therefore, SEI thickness could be lower than 315 nm if some of parasitic reactions form gaseous. This could be discussed and clarified in the text.
- The authors write that "the choice of 10 μ Ah charge steps (at 5 mA) led to short-circuiting of some cells, and therefore, smaller charge steps or alternative counter electrodes (e.g. In/InLi) are recommended for high current/charge step tests". However, in this study, only Li plating was applied on CC (no stripping). Why should the choice of the counter electrode effect the short-circuiting? Isn't that the high charge and current leads to formation of dendrite on working electrode leading to short circuits?

Detailed responses to reviewer comments for Nature Communications

Manuscript ID: NCOMMS-23-32284

*We thank the reviewers for their feedback and valuable time. We believe that their feedback helped us to improve the manuscript –which is highly appreciated. Changes made in the revised manuscript are highlighted in green. We hereby give point-by-point answers to their comments. Our responses will be given in **italic/blue** font as in this text.*

Reviewer #1 (Remarks to the Author):

Aktekin et al. reported a new electrochemical method, called “Coulometric Titration Time Analysis (CTTA)” to quantify the SEI formation at the interface of Li metal and sulfide electrolytes. This method not only performs a rarely reported technique in quantifying SEI formation of side reactions by calculating the accumulated charge, but also shows the potential to analyse the issue of lithium dendrites. CTTA can also be used to explore a wide range of different electrolytes and current collectors as a function of all relevant parameters. ***The reviewer has acknowledged the high novelty and potential generalizability of this study and suggests publishing after minor revisions.*** Specific comments are outlined below:

1. According to several important review papers involving the interface characterization on sulfide-based all-solid-state batteries (e.g., doi.org/10.1007/s41918-022-00176-0; doi.org/10.1002/sstr.202100146; DOI: 10.1039/D1EE00551K), the interface issue on Li metal/sulfide includes two basic aspects, namely interfacial side reaction and Li dendrites. The authors quantified the SEI formation by calculating the accumulated charge, but how can this reflect the specific contribution from side reactions or Li dendrites, respectively? Or must be combined with other characterization techniques to give this insight?

Thanks for the very positive report!

We agree that interfacial side reactions and formation of lithium dendrites are two important issues regarding the use of lithium metal with sulfide-based SEs. Electrochemical methods which are commonly used to quantify the degree of side reactions are either based on more static experiments such as CV (and LSV) tests, or more dynamic experiments such as Coulomb efficiency (CE) measurements. While experiments such as CV/LSV can allow us to acquire information from side reactions only, it is not possible to account for the active electrode material's impact on side reactions. On the other hand, CE measurements can account for active material, but CE will be affected by both side reactions and dendritic lithium growth effects (e.g. dead lithium formation, pronounced kinetic effects, etc.). The CTTA technique deals with this problem by plating only a small amount of lithium at rather low currents in each titration step; and subsequent plating is only done once the previously plated lithium is consumed by side reactions. These conditions are not in favor of dendritic lithium growth, and due to geometrical considerations, the possibility of dead lithium formation is rather low. This is corroborated by the observation that at 1 μAh step charge we did not see any effect of increasing the current to 5 mA ($\approx 7.5 \text{ mA cm}^{-2}$) from 10 μA ($\approx 15 \mu\text{A cm}^{-2}$), as shown in Figure 4a.

On the other hand, by increasing both the current and the amount of lithium plated in each titration step, it is possible to promote dendritic lithium growth. This could make an additional contribution to the accumulated charge values observed in the experiment, and therefore provide useful information in determining the stability/performance of different electrolytes against dendritic lithium growth. The difference in accumulated charge between the low current/low charge and high current/high charge experiments may be attributed to the dendritic growth, however, it may not simply be used to quantify dead lithium formation since the degree of side reactions would also be increased due to higher surface area of lithium, and also penetration of lithium into fresh electrolyte regions. Therefore, quantification of SEI growth is proposed for low current/low charge titration conditions with this technique, and high current/high charge conditions are only proposed to get additional information on the dendritic lithium growth effects (qualitatively). We believe that the combination of this technique with other post mortem characterization techniques such as titration gas chromatography or mass spectrometry titration will allow better quantitative understanding of dendritic growth in future studies.

We made the following changes in the revised manuscript.

Page 2. “... This effect could be exacerbated during lithium plating due to volume change and SEI damage associated with the freshly deposited lithium metal, especially if dendritic lithium growth occurs.³⁵ Therefore, it is important to complement such voltammetric tests with other analytical methods.^{36,37}”

Page 7. “... thereby causing additional accumulated charge in Figure 4-b (it should be noted that separating these two processes would require additional post mortem analysis). A lower cell resistance is also observed initially, however, as also observed in Figure S1, ...”

2. Consider limitations with severe interfacial side reactions: If severe chemical reactions occur between Li metal and certain SEs (e.g., metal-containing halide SEs), the kinetics of the interface reaction may not stabilize. Address this limitation of the CTTA method and its applicability under such conditions in the main text.

This is a quite good question. We believe that this technique can be quite useful in identifying electrolytes which do not form a self-limiting SEI, such as metal-containing halide SEs. For instance, if an SEI with high electronic conductivity is forming (i.e., an MCI in our original notation, see Wenzel et al., Solid State Ionics 278, 2015, 98), the consumption rate of lithium would be relatively fast and not likely follow a parabolic charge accumulation trend. The representation of results with square-root of time in x-axis would show deviations from linearity easily.

The comparison of different electrolytes in Figure 5 in the manuscript show the practical relevance by comparing LGPS to LPS and LPSCI – and is a partial answer to the reviewer’s question. In terms of quantification vs. time, in case of a highly reactive electrolyte, the degree of side reactions may be so high so that the titrated lithium can be consumed simultaneously during lithium plating (e.g. see Figure 4-d for the aluminium current collector). In such a case, the potentiostat can limit the current to the value which was pre-determined in the program by acting as a ‘brake’, and therefore the accumulated charge vs. time trends can show a more optimistic scenario than the actual behaviour; however, this problem can be overcome easily by increasing the titration current. However, even under low current

conditions, the technique will already show that the electrolyte is not suitable to be used in a practical battery cell system and therefore it would not be necessary to further study such an electrolyte in any case. Therefore, the experiment with an Al current collector is a good example of this case. A less severe case is also shown in Figure 5; here the degree of side reactions for LGPS is already beyond acceptable limits for any practical battery system – as well known.

We made the following changes in the revised manuscript.

Page 8. “... *In fact, lithium is consumed simultaneously during the coulometric titration steps rendering the experiment simply to a constant-current discharge procedure (which could be avoided if higher titration currents were chosen).*”

3. Discuss whether the CTTA method needs to account for joule heat caused by the resistance of the test cell, as it could potentially influence the calculation of the accumulated charge.

As explained in our respond to comment 1, titration conditions for lithium plating are chosen in a way to plate a small amount of lithium in each titration step, and there is a considerable pause time (i.e. OCV) between each titration step. Therefore, we do not foresee a significant contribution due to Joule heating. If the technique is to be used under harsh conditions in the future to study dendritic lithium growth, especially if the lithium plating has preferential spots on the current collector; we agree with the reviewer that the consideration of Joule heating would be important for the interpretation of results. However, it should be noted that even under such harsh conditions, still the titration conditions need to be adjusted in a way to not allow too much lithium plating since the main idea of the technique is to plate lithium which then is consumed by side reactions in reasonable time, unless highly reactive SEs are tested. Therefore, such effects would still be less critical as compared to practical cycling conditions and CE measurements.

We made the following changes in the revised manuscript.

Page 8. “*It is also important to consider thermal effects, e.g. local temperature variations due to Joule heating, particularly at high charge steps / high currents.*”

4. There is a large length on the interface characterization for the Li/LPSC, which might be deviated from the key point of this paper. Also, a lot of similar work have been published to explain the interface between Li metal and LPSC. Therefore, the reviewer suggests to re-arrange this part to make the paper more readable.

The main aim of this paper is to demonstrate a new electrochemical approach to the scientific community, and we totally understand why the reviewer makes this suggestion to reduce the length of the manuscript in a way to make the paper more focused on the technique itself. This was also a point we considered many times during the preparation of the manuscript. However, eventually we decided to keep it in its current format for several reasons as briefly explained below.

Firstly, we believe that the morphological characterization of the current collector and SE surfaces are important for this work and needs to be in the main text as it corroborates the SEI thickness estimations from the CTTA results after 400 hours, but also shows that the

lithium deposition and SEI growth are heterogeneous, the latter particularly being the case during the early stages (Figure 6 and S5). These morphological characteristics of SEI films also make it important to understand the occurrence of side reactions on different electrode areas. For this reason, we investigated the formation of local “galvanic cells” where certain areas of the working electrode can act as anode and other areas as (quasi) cathode. This is presented in Figure 7. In Figure 8 and Figure 9, we present the SEI characteristics as obtained by XPS and ToF-SIMS techniques. Investigations with these techniques have been reported in other earlier studies, however, we believe it is important to report them in the main text (please note that we tried our best to keep the length of discussion brief) as they provide an important cross-check for the validity of the CTTA technique. It should be noted we already moved a significant portion of XPS and ToF-SIMS results to the supplementary information. Also, the role of oxygen containing compounds in the SEI growth has been overlooked in the past and we make important observations on this aspect. The latter is also important for the readers to understand since the involvement of oxygen could also contribute to the charge accumulated in the CTTA experiments.

We felt that we need to properly justify all features of the CTTA technique as good as possible, to convince the readers. In the lights of these reasons, we hope that the reviewer understands our concerns and why we decided not to shorten the mentioned sections in the manuscript.

5. Attend to non-scientific issues: Ensure consistency in reference style, provide missing page/article numbers for references (e.g., refs. 65, 69, 70, 74), and maintain uniform abbreviation of journal names. Additionally, indicate the value of stack pressure in all relevant figures.

We thank the reviewer for their careful reading of the manuscript. We made the corrections in the manuscript (highlighted in green in the revised manuscript).

Reviewer #2 (Remarks to the Author):

This paper uses a novel electrochemical method to quantify the amount of side reactions of a solid-state electrolyte, especially LPSCI. It does an in-depth investigation within the field of electrochemistry and aims to measure the quantity of actual side reactions in solid-state electrolytes by systematically eliminating various other possibilities in their electrochemical analysis. Although the length of this paper may be not relatively concise to be published in Nature Communications and focuses much on the electrochemical analysis which might be not the best way to appeal to a broad audience, ***I believe that it is still considered sufficient for publication in this journal if they address these minor comments.***

Thanks for the generally positive report!

1. The authors claimed to quantitatively determine the amount of side reactions in solid-state electrolytes by counting accumulated charge. However, a precise explanation is needed for how the absolute quantity of charge consumed for other reasons can be excluded. The authors

already mentioned the possibility of the charge consumed such as during the initial step when dropping from 2.5V to 0V and the charge consumed due to the native oxide layer in the CC. However, there seems to be an insufficient explanation of how to subtract these quantities to calculate the charge consumed during actual side reactions in solid-state electrolytes only.

Thanks for this helpful question, which shows that we still can improve the presentation of the technique. We agree with the reviewer that different processes can contribute to the CTTA signal, beyond the mere electrolyte reduction reactions of the SE. We assume that all other contributions will have a minor contribution compared to LPSCl reduction, and we demonstrate that this assumption is correct by performing a number of additional experiments.

*If a solid electrolyte which is known to be completely stable at the chemical potential of lithium is used as SE, the CTTA results would only reflect the sum of all other contributions such as native oxide decomposition of current collector, diffusion loss into the current collector, or reduction of impurities present in/on cell components, etc. Fortunately, LLZO SE is known to be stable at the low potential of lithium metal, and therefore we used this material as SE in a reference CTTA experiment in order to determine the sum of these contributions apart from the LPSCl decomposition. In Figure 5, we show the results from a similar cell prepared with LLZO SE. This cell shows a charge accumulation of about 4 μAh after 400 hours (where most of this charge was already consumed within the first 100 hours). This is a minor contribution as compared to the total charge accumulated in LPSCl-based cells. As the LLZO stability is also under debate, there could also be some contribution from LLZO reduction to this value of 4 μAh , but this would mean that the contribution of other processes would be even lower in this case. Apart from this, a significant portion of this charge is likely originating from the oxygen impurities in/on the cell parts and SE, and cause formation of compounds such as Li_2O , LiOH , etc., which can be accepted as part of SEI-films as commonly reported in the literature. However, we agree with the reviewer that a more explicit explanation/discussion will be useful **and therefore we added the following sentence** in the revised manuscript.*

Page 9. *“... The accumulated charge results of LLZO-based cells can also be subtracted from the results of cells with different SEs (tested under identical conditions) in order to obtain a quantitative estimate of side reactions caused solely by the SE reduction reactions.”*

2. Figures 2d, 2e, and others seem to present the results of quantifying the side reactions in solid-state electrolytes using the claimed CTTA method. However, the absolute amount to which the solid-state electrolyte has decomposed is not thoroughly discussed. It would be essential to provide more detailed information, such as the moles or thickness of the solid-state electrolyte that underwent side reactions, through figures or tables as this would be a key result of this analysis.

We apologize if we have not been explicit enough with this issue. Indeed we estimate SEI thickness. As discussed in the previous section, the amount of accumulated charge indeed largely corresponds to the SE reduction reactions. However, in SEI thickness estimations (and therefore estimation of moles of LPSCl reacted, or moles of Li_2S , LiCl , Li_3P formed), it is not possible to account for the contribution of each additional side reaction, and therefore we assumed solely LPSCl reduction. This is also the case with the assumption of non-porous and

homogeneous planar SEI-film formation which is already discussed in the manuscript as films are in practice heterogeneous and porous. Therefore, these estimations are indeed made to give a rough estimate on the length scale of SEI films (note that these do not compromise the accuracy of lithium consumption quantification). Even though we already discussed these issues in the manuscript, we agree with the reviewer that these points should be more clearly stated in the manuscript, especially when the SEI-thickness estimates are made. Therefore, we added a sentence on Page 6 as shown below. We also added a table in which we show the details of the moles and volumes of reactants and products involved in each titration step, i.e. 1 μAh charge.

Page 6. “... a rough estimate of the SEI thickness (see Table S1 for the detailed information on the moles and volumes of reactants and products) can be made assuming the decomposition reaction ...”.

Table S1. Estimation of number of moles of $\text{Li}_6\text{PS}_5\text{Cl}$ and lithium metal reacted during each titration step (1 μAh) in standard CTTA experiments. Thickness estimations for each reactant and product are also given for 1 μAh lithium titration steps. Faraday’s constant is taken as 26800 mAh, and density values were retrieved from the Materials Project.¹ It should be noted that these estimations assume i) the reactants and reaction products given in the table, ii) consumption of titrated lithium completely in the given reaction, iii) formation of a dense and homogeneous SEI film consisting of given reaction products in the table. As shown in the manuscript via electrochemical experiments and post mortem characterization of SEI-films, additional side reactions can also make a minor contribution to the overall lithium consumption (reaction of lithium with contaminations in/on cell components or thin native oxide film on the current collector, or the current collector itself, etc.), porous and heterogeneous SEI-films can form, and compounds not given in the proposed reaction can be present in the SEI-films such as Li_2O or LiOH . Due to these reasons, SEI thickness estimates should only be used to get an approximate information on the length-scale of the SEI-films forming after a given time. We assume that these approximate thicknesses are quite reliable in most cases, if the current collector is chosen well and if samples are prepared properly.

-	n (mol)	Compound	n (mol)	Comp.	→	n (mol)	Comp.	n (mol)	Comp.	n (mol)	Comp.
Reaction	1	$\text{Li}_6\text{PS}_5\text{Cl}$	8	Li	→	5	Li_2S	1	LiCl	1	Li_3P
Density (g cm^{-3})		1.64		0.56	→		1.67		2.14		1.48
Molar mass (g mol^{-1})		268.4		6.9	→		46		42.4		51.8
Molar vol. ($\text{cm}^3 \text{mol}^{-1}$)		163.7		12.3	→		27.5		19.8		35
n * Molar vol.		163.7		98.6	→		137.7		19.8		35
1 μAh charge (e.g. Li^+)	$4.7 * 10^{-9}$ mol of	$\text{Li}_6\text{PS}_5\text{Cl}$	$3.73 * 10^{-8}$ mol of	Li	→	$2.3 * 10^{-8}$ mol of	Li_2S	$4.7 * 10^{-9}$ mol of	LiCl	$4.7 * 10^{-9}$ mol of	Li_3P
Volume ($\text{cm}^3 \mu\text{Ah}^{-1}$)		$7.6 * 10^{-7}$		$4.6 * 10^{-7}$	→		$6.4 * 10^{-7}$		$9.2 * 10^{-8}$		$1.6 * 10^{-7}$
Thickness ($\text{nm } \mu\text{Ah}^{-1}$)		7.6		4.6	→		6.4		0.9		1.6
Thickness ($\text{nm } \mu\text{Ah}^{-1}$)	12.2				→	9.0					

Ref.1. The Materials Project: A materials genome approach to accelerating materials innovation, Anubhav Jain, Shyue Ping Ong, Geoffroy Hautier, Wei Chen, William Davidson Richards, Stephen Dacek, Shreyas Cholia, Dan Gunter, David Skinner, Gerbrand Ceder, and Kristin A. Persson.

3. After plating/undergoing side reactions under the applied pressure conditions and the cell disassembled for SEM analysis of LPSCl or CC surfaces, how do the SEM images have

spherical deposit morphologies? How does it maintain a spherical shape rather than a distorted shape?

Even though the formation of spherical particles resembles the initial lithium deposition sites, we show that these particles are also formed in external short-circuiting experiments and consist of oxygen-rich cores, indicating that they are formed during SEI formation. We considered the possibility of their formation during sample transfer to the SEM and ToF-SIMS instruments, or during the imaging. To test this possibility, we scanned certain areas for long durations, but we did not observe formation of these species. On the other hand, we opened the cells in a glovebox and prepared the samples for characterization immediately after the cell opening. As can be seen in below SEM images, we observed some particles which were mechanically fractured during sample preparation (e.g. during removal of stainless steel foil from SE pellet) resulting in detachment of particles partially to the either side. This provides strong indication that these particles were formed prior to sample transfer, likely during SEI formation. How they maintain the spherical morphology under a constant stack pressure of 13 MPa is not clear to us, but we believe that it is due to the heterogeneous microstructure of SEI films (presence of different compounds with different mechanical properties and porous microstructure) and surface roughness of SE pellet surface. Due to these, local pressure can vary considerably and may enable formation of spherically shaped particles.

We added the below SEM image into SI and made the following changes in the revised manuscript.

We made the following changes in the revised manuscript.

Page 9. “... Some of these particles were fractured during the removal of the current collector (which was done directly after cell opening in the glovebox) indicating that these particles were formed during the experiment which was performed under 13 MPa stack pressure (see Figure S5)”.

4. Regarding the claim of P-deficient SEI formation, what is the own evidence and reasoning presented in this paper, apart from the previous literature's opinion? Is it merely because LPSCI contains 5 S atoms per 1 P atom, or are there other reasons for this claim?

This is an important question, but the exact reason for this behaviour is not clear, and we believe this should be explored in more depth in future studies. However, it is not due to

different number of atoms for S, P, and Cl in the LPSCl stoichiometry. For instance, as given in Figure 8, we calculated the atomic fractions of elements in the analysis area (before and after surface cleaning with argon sputtering) for both SE-side and current collector-side. As expected from the stoichiometry, Cl and P percentages are quite similar for the pristine SE pellet (e.g. see the trends after surface cleaning). After SEI formation, the relative ratio of P to Cl changes significantly as the SEI becomes rather P-deficient, probably due to diffusion of P-containing SEI products from the Li₂S-rich SEI. We can only speculate that the possible SEI product such as Li₃P may further react with cell contaminants and form volatile species since we did not observe any other region which is rich in P-compounds. But this explanation is a speculation at this point, and it requires future experimental and computational investigations.

5. Is there any method available to quantify oxidative side reactions of solid-state electrolytes as well as reduction reactions?

As in the case of quantification of reductive side reactions, both CV/LSV, step-wise CV, and CE measurements have some limitations. We considered about the possibility of performing CTTA experiments for the cathode side, however, such an experiment could also have its own limitations. In CTTA technique, one important requirement is that the working electrode should change its potential abruptly with the application of a small amount of charge and it should not be prone to kinetic limitations. Most cathode active materials (CAMs) do show a sloppy voltage profile and transport kinetics within the material is not sufficiently fast. Nevertheless, the application of the technique could be possible with some CAMs which may show an abrupt change in its electrode potential upon delithiation, and which has a high ionic conductivity for lithium ions.

Reviewer #3 (Remarks to the Author):

The manuscript NCOMMS-23-32284 by Prof. Janek et al. discusses new method on quantifying SEI on lithium metal anode in solid state batteries. The manuscript investigates the influence of number of parameters such current density, plated capacity, temperature, and composition of the substrate on the amount of SEI formed on lithium metal with the proposed titration method. ***The work includes interesting results and discussion, so I think it fits to Nature Communications, however there are some comments below to be addressed:***

Thank you for this very positive report!

1. It is not completely clear how the “accumulated charge” was calculated in this work. Please provide more details in the beginning of the results section to clarify it.

We added more details of this calculation in the relevant part of the revised manuscript as suggested by the reviewer.

Page 6. “... The amount of accumulated charge consumed in side reactions ($q_{\Sigma} = \Sigma q(\tau_i) = \text{number of titrations times step charge}$) with respect to the duration of the experiment...”.

2. The initial OCV before starting the experiment was 2.3-2.5 V when stainless-steel was used which is assigned to minor sulfur redox. Could the author add information on the initial OCV with the other tested current collectors? How could the initial OCV be explained with the other current collectors (i.e. Al, Cu, Ni)? Any comments on the influence of electrolyte impurities on the initial OCV?

We checked the data for these cells, and the open circuit potentials of these cells are as following. OCV-Al \approx 1.93 V vs. Li⁺/Li, OCV-Cu \approx 1.83 V vs. Li⁺/Li, OCV-Ni \approx 2.35 V vs. Li⁺/Li.

The initial OCV depends sensibly on the experimental conditions and materials history. An OCV is well defined and also stable, as long as one uses two reversible (and non-polarizable) electrodes. This is e.g. the case directly after titration, when both electrode potentials are well defined by the Li⁺/Li redox couple. Before the first titration, the redox equilibrium at the (lithium metal-free) working electrode, i.e., the current collector, depends very much on the equilibration time of the cell before any measurement and any impurity at the interface. In the ideal case of a clean interface and long equilibration, the SE at the current collector will show the same chemical potential of lithium metal within the SE as at the counter electrode – however, without the presence of lithium metal as a real phase. Thus, the potential of the working electrode will be defined by the only charge carrier that can equilibrate, and this is the electrons in the SE. Now, as electrons are the minority charge in the SE, this electronic equilibrium will depend sensibly on the experimental conditions. While the OCV before the first titration step contains information on the initial redox state of the working electrode interface, this information is not easily interpreted. We take the question of the reviewer as trigger to think deeper about this point. However, for the CTTA method, both the initial OCV as well as the OCV after consumption of lithium after titration are not relevant quantitatively. The method relies simply on the fact, that the OCV will take non-zero values once lithium metal is not present at the current collector.

3. The assumption of formation of SEI with thickness of almost 315 nm is based on that all the side products contribute to formation of SEI. However, another important mechanism is formation of gaseous by side reaction. Therefore, SEI thickness could be lower than 315 nm if some of parasitic reactions form gaseous. This could be discussed and clarified in the text.

*We agree with the reviewer. In SEI thickness estimations, we assumed formation of a compact and homogeneous SEI film consisting of Li₂S, LiCl and Li₃P. This estimation has limitations and therefore **we added a clarification** on this issue in the revised manuscript.*

Page 6. “... would result in an SEI thickness $d \approx 9$ nm (assuming a compact mixture of Li₂S, LiCl and Li₃P, and absence of gaseous products). This would correspond to an estimated SEI thickness of $d \approx 315$ nm after one week and $d \approx 540$ nm after about 400 h (~17 days).”.

4. The authors write that “the choice of 10 μ Ah charge steps (at 5 mA) led to short-circuiting of some cells, and therefore, smaller charge steps or alternative counter electrodes (e.g. In/InLi) are recommended for high current/charge step tests”. However, in this study, only Li plating was applied on CC (no stripping). Why should the choice of the counter electrode

effect the short-circuiting? Isn't that the high charge and current leads to formation of dendrite on working electrode leading to short circuits?

At the beginning of this project, we also assumed that the CE would not have an effect in such a study since the experiment mainly gives information on the current collector side (WE). However, particularly at high currents (e.g. 5 mA in this study), we observed the occurrence of short-circuits in some cells. This observation could not be made when we used In/InLi electrodes in additional experiments. We believe that the current focusing at certain spots on the counter electrode at 5 mA current – due to void formation or insufficient contact due to contaminations on the lithium metal (such as Li_2O , LiOH , Li_2CO_3) affects the lithium plating characteristics indirectly on the WE. If nuclei density on the WE are affected by the current localization on the CE, this could also have an impact on the possibility of short-circuit occurrence. This observation will be investigated in more detail in a future study.

We are grateful to all three reviewers for their careful reading of the manuscript and for their valuable comments.

Reviewers' Comments:

Reviewer #1:

Remarks to the Author:

The authors have addressed all comments and suggestions well. It is publishable now.

Reviewer #2:

Remarks to the Author:

The author has addressed the comments from the reviewer sufficiently, so I believe this paper can be published in this journal.

Reviewer #3:

Remarks to the Author:

The authors have properly revised the manuscript, so it can be published in Nature Communication.